# Geomorphic indicators of continental-scale landscape transience in the Hengduan Mountains, SE Tibet, China

Katrina D. Gelwick[1], Sean D. Willett[1], Rong Yang[2]

[1]Department of Earth Sciences, ETH Zürich, Zurich, 8092, Switzerland
[2]School of Earth Sciences, Zhejiang University, Hangzhou, 310027, China

*Correspondence to*: Katrina D. Gelwick (kgelwick@ethz.ch)

**Abstract.**

Landscapes are sculpted by a complex response of surface processes to external forcings, such as climate and tectonics. Several major river captures have been documented in the Hengduan Mountains, leading to the hypothesis that the region experiences
exceptionally high rates of drainage reorganization driven by horizontal shortening and propagating uplift. Here we determine the prevalence, intensity, and spatial patterns of ongoing drainage reorganization in the Hengduan Mountains and evaluate the relative time scales of this transience by comparing drainage divide asymmetry for four geomorphic metrics that operate at different spatial and temporal scales. Specifically, we calculate the migration direction and the divide asymmetry index (DAI) drainage divide asymmetry in catchment-restricted topographic relief (CRR), hillslope gradient (HSG), normalized channel
steepness ($k_{sn}$), and normalized channel distance ($\chi$). $k_{sn}$ and $\chi$ are both precipitation-corrected to account for the strong precipitation gradient across the region. The different spatial scales of these geomorphic metrics allow us to establish the relative timescales of observed landscape transience in the Hengduan Mountains, where local scale metrics measure short-term change and integral quantities measure long-term disequilibrium. We find a high incidence of strongly asymmetric divides in all metrics across the Hengduan Mountain region. Although the magnitude of asymmetry varies significantly between
metrics, possibly due to a combination of metric-specific thresholds and varying proxy relationships with erosion rate, a majority of divides agree on divide migration direction. Agreement in divide migration direction indicates active landscape response when asymmetry is high and a state of quasi-equilibrium when asymmetry is low. Disagreements between the integral quantity, $\chi$, and the other geomorphic metrics can be explained by different timescales of the underlying geomorphic processes, with $\chi$ reflecting a long-term response and CRR, HSG, and $k_{sn}$ capturing short-term perturbations to catchment structure. These
perturbations include various transient mechanisms, such as differential tectonic uplift or erodibility, glacial alteration, and river captures. Our work confirms the high incidence of drainage reorganization across the Hengduan Mountains and highlights both transient and stable areas in the landscape with high resolution. We also offer valuable insights on the application of geomorphic metrics that can be generalized and applied to the study of landscape transience and drainage divide asymmetry in other settings.

# 1 Introduction

The Earth's landscapes are constantly evolving as a result of the complex interplay between surface processes and external forcing, such as climate and tectonics. Landscape change occurs through both geometric transience, which we use to refer to planform changes in the drainage network, and topographic transience, which refers to changes in relief or elevation. Planform drainage network geometric (henceforth "geometric") and topographic transience are closely linked, as changes in drainage patterns can have a significant impact on the relief and elevation of the landscape and, reciprocally, changes in topography can trigger changes to the drainage network.

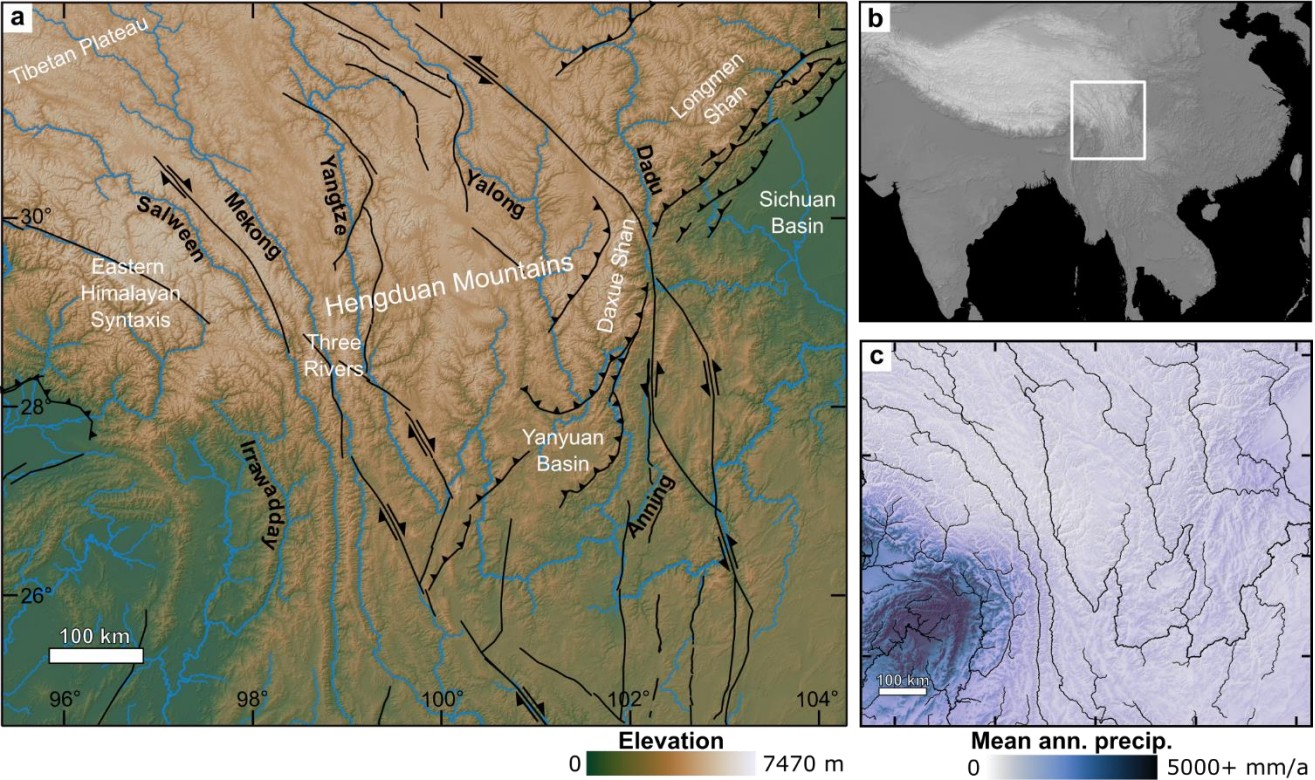

Figure 1: (a) Elevation map of HDM region showing major faults in black. Rivers are shown in blue and labelled in black. White labels denote key areas mentioned in text. (b) Topographic map of Central Asia; white inset box shows extent of a & c. (c) Map of mean annual precipitation in the HDM (Karger et al., 2017, 2018). Rivers are shown in black.

A prime example of this connection between geometric network change and topography is observed in the high-elevation, low-relief areas scattered throughout the Hengduan Mountains (HDM), Southeast Tibet. Many of these low-relief features have been interpreted to result from river capture, where drainage area loss inhibits the ability of catchment erosion to keep pace with background uplift (Yang et al., 2015; Willett, 2017; Fox et al., 2020; Yuan et al., 2021). This hypothesis is supported by several major river captures in the HDM which indicate significant, ongoing drainage reorganization in the region (Clark et al., 2004; Zheng et al., 2021). While few of these captures have been decisively dated, several have been confirmed

or estimated to have occurred since 2-4 Ma (Kong et al., 2012; Liu et al., 2020; Sun et al., 2020; Yang et al., 2020). Another explanation for these low-relief features is a delayed incisional response in small tributaries to propagating tectonic uplift from the ongoing India-Eurasia collision (Clark et al., 2006; Whipple et al., 2017a, b). Glacial planation may have also played a role

in their formation in previously glaciated areas (Zhang et al., 2016). Identifying transients in the river network can help to diagnose the origins of low-relief features in the HDM and distinguish between these hypotheses (Whipple et al., 2017a, b; Willett, 2017; Fox et al., 2020) . Despite its critical role in shaping the landscape, the prevalence, intensity, and spatial distribution of geometric transience has not been systematically measured across the HDM on a large scale.

      The geometry of a drainage network moves towards a quasi-topographic equilibrium as the system adjusts to spatial

asymmetries in erosion and uplift (Beeson et al., 2017). This happens through drainage divide migration, which can occur by both continuous shift in the position of a divide and discrete river capture events. As the direct measurement of catchment erosion rates is difficult, previous studies rely primarily on across-divide differences in geomorphic steepness metrics as a proxy for differential erosion to infer active drainage exchange between catchments. Common metrics include mean hillslope gradient, mean local relief, stream channel steepness, channel head elevation, and hillslope curvature measured near the divide

(Hurst et al., 2013; Whipple et al., 2017c; Forte and Whipple, 2018; Scherler and Schwanghart, 2020; Zhou et al., 2022b).

      In addition to these local-scale metrics, $\chi$, a transformed variable of the along-stream distance (Perron and Royden, 2013), has been widely applied to assess the general geometric stability of the drainage network pattern with the assumption that planform patterns in $\chi$ should be reflected in the distribution of divide elevation and symmetry (Willett et al., 2014; Beeson et al., 2017; Sassolas-Serrayet et al., 2019; Ye et al., 2022; Zhou et al., 2022b). However, the direct interpretation of $\chi$ as a

proxy for steady-state divide elevation requires that rock uplift and erodibility have the same mean values, integrated along the channel (Perron and Royden, 2013; Willett et al., 2014), which is unlikely to hold over the scale of entire mountain ranges at any given time. While $\chi$ is easily adjusted to account for variable erodibility due to spatial patterns in precipitation (Yang et al., 2015), other effects, such as lithology and local variations in uplift, remain challenging to incorporate (Wu et al., 2022; Zhou et al., 2022a; Dal Pai et al., 2023). In addition, elevation transients in the stream network, e.g., from a change in uplift

rate, require time to propagate upstream to water divides. As $\chi$ represents the equilibrium elevation and not the instantaneous elevation, it will not accurately reflect true channel head elevation or steepness during elevation transients. Metrics which reflect local erosion and uplift dynamics are thus more reliable predictors of instantaneous motion of specific drainage divides at a specific time (Whipple et al., 2017c; Sassolas-Serrayet et al., 2019; Dal Pai et al., 2023). However, $\chi$ may still indicate future divide instabilities within a catchment, even in landscapes with variable erodibility and uplift (Forte and Whipple, 2018),

as eventually uplift rates and erodibility are likely to equilibrate and transients will propagate through the network.

      Local-scale metrics are also subject to variations in local physical properties and transients and regularly exhibit large variability along drainage divides, as well as internal contradictions between metrics (Sassolas-Serrayet et al., 2019; Dal Pai et al., 2023). To mitigate this, studies often combine multiple metrics and/or take the mean value of all catchments along each side of a main drainage divide (Forte and Whipple, 2018; Sassolas-Serrayet et al., 2019; Zhou et al., 2022b; Dal Pai et al.,

2023). However, the underlying mechanisms responsible for variability within or between metrics that measure local or

regionally integrated quantities are likely to be different and resolution of this variability requires a systematic evaluation in transient landscapes.

This study aims to determine the prevalence, intensity, and spatial patterns of ongoing geometric transience in the HDM by evaluating drainage divide asymmetry using four geomorphic metrics that operate at different spatial and, consequently, temporal scales. Our approach enables us to differentiate between transient and relatively stable sections of the HDM drainage network and deepen our understanding of the mechanisms, drivers, and relative time scales of landscape evolution in this dynamic region. We also take the opportunity to compare the internal self-consistency of geomorphic metrics in predictions of drainage divide migration across the tectonically and climatically diverse HDM (Fig. 1). We examine the degree of agreement between metrics on the direction of divide migration and the impact of asymmetry magnitude on that agreement. The particular case of spatial disagreement between $\chi$ and local-scale metrics is investigated in more depth to determine whether disagreement is due to violation of the assumption of constant mean uplift and erodibility, or if it reflects cases where long-term transience has not yet propagated through the catchment to reach the divide.

## 2 Background

The HDM are a series of northwest-southeast oriented mountain ranges extending from the south-eastern margin of the Tibetan Plateau (Fig. 1a,b). The region encompasses ~600,000 km$^2$ and has highly variable relief, with elevations ranging from less than 300 m to over 5,000 m. The highest peak, Gongga Shan, reaches 7,470 m and lies within the Daxue Shan range on the HDM's eastern flank.

### 2.1 Drainage Network

The HDM are dissected by several major rivers, most notably the Yangtze (Jinsha), Mekong (Lancang), and Salween (Nu), whose headwaters originate on the Tibetan Plateau (Fig. 1a). These three rivers form steep gorges which run parallel to each other for over 1,000 km (Fig. 1a, "Three Rivers"), before carving divergent paths across the landscape and eventually draining to the East China, South China, and Andaman seas, respectively. River capture has significantly altered the drainage network in the HDM (Clark et al., 2004), however the timing and drivers of landscape transience are still debated, with hypotheses including the onset of regional surface uplift and strike-slip faulting in the Eocene (Clark et al., 2004; Gourbet et al., 2017), an intensified monsoon in the Miocene (Zheng et al., 2021), volcanic activity (Gourbet et al., 2017), and/or an increase in localized uplift rate in the Plio-Pleistocene (Yang et al., 2020). Documented major captures include the rerouting of the upper Yangtze from the paleo-Red River sometime between the Eocene and Pleistocene (Kong et al., 2012; Sun et al., 2020; Liu et al., 2020; Zheng et al., 2021) and the diversion of the upper Dadu from its previous path through the Anning River in the Pleistocene (Yang et al., 2020). Many rivers in the HDM align with fault traces for long stretches (e.g., Red, Yalong, and Anning rivers), making the drainage network particularly susceptible to tectonic activity.

## 2.2 Tectonics and Lithology

The HDM are framed by the rigid crustal block of the Sichuan Basin to the east and the Burma Block to the southwest (Wang et al., 2017a). The region is dominated by large NE-SW trending strike-slip faults (Fig. 1a), with dextral faults on its southwest margin (e.g., Red River Fault) and sinistral faults on its northeast margin (e.g., Xianshuihe-Xiaojiang fault system). Faults with a predominantly thrust component, including the Yalong Thrust Belt, run roughly perpendicular in a SW-NE orientation, in line with the neighboring Longmen Shan fault system. Active movement is also documented on normal faults in the HDM, including a 6.6 Mw earthquake near the First Bend of the Yangtze River in 1996 (Ji et al., 2017).

The HDM has experienced active uplift and faulting since at least the Eocene (Su et al., 2019) with additional localized increases in uplift rates observed in the Plio-Pleistocene (Li et al., 2018). The region's strike-slip faults are thought to accommodate part of the tectonic strain from the ongoing India-Eurasia collision through the extrusion and rotation of material from the Tibetan Plateau around the Eastern Himalayan syntaxis and through the HDM (Tapponnier et al., 1982; England and Houseman, 1986; Leloup et al., 1995; Tian et al., 2014). Alternative models for the deformation and uplift of the HDM include crustal thickening (Kirby and Ouimet, 2011) and lower crustal flow (Royden et al., 1997; Clark et al., 2004, 2006). The role of early topography and river reorganization due to the accretion of the Burma terrane is also potentially important, but not well investigated (Westerweel et al., 2019).

The complex tectonic history of the HDM has resulted in a heterogeneous surface geology (Fig. S1). Bedrock in the region predominantly consists of sedimentary formations, ranging from Paleozoic to Eocene in age, and Pre-Mesozoic metamorphic rocks, along with some Mesozoic and Cenozoic granites (China Geological Survey Bureau, 2019; Hartmann and Moosdorf, 2012b, a). Neogene and Quaternary deposits are found in many sedimentary basins and low-relief river catchments (e.g., Yanyuan Basin, Anning River).

## 2.3 Climate

The HDM are largely temperate in climate, transitioning to subtropical in the south and becoming increasingly cold and arid in the northwest as they merge into the Tibetan Plateau (Kottek et al., 2006). Precipitation in the HDM varies seasonally due to its position at the nexus of the Asian Monsoon (both South Asian and East Asian) and the Westerlies, which drive wet summers and dry winters, respectively (Wang et al., 2010). Annual precipitation in the region varies by over an order of magnitude, ranging from approximately 180-3,260 mm/a, with most precipitation concentrated in the west due to a combination of the South Asian Monsoon and topographic effects (Bookhagen and Burbank, 2010; Karger et al., 2017, 2018; Fig. 1c). Another, weaker summer precipitation high in the eastern HDM is controlled by the East Asian Monsoon. Monsoon onset is debated, but the region has experienced a strong monsoonal influence since ~20 Ma (Betzler et al., 2016; Tada et al., 2016).

During the last glacial maximum ~20 ka, glaciers in the HDM extended below at least 4,000 m (Fu et al., 2013). Residual evidence of Pleistocene glaciations is found throughout the HDM in the form of moraines, glacial lakes, and other

landscape features (Fu et al., 2013; Zhang et al., 2016). Glaciers persist in some of the high elevation (>5,000 m) portions of the HDM, particularly in the Three Rivers area and Daxue Shan (Wang et al., 2017b).

## 3 Methods

### 3.1 Data Sources

All geomorphic analyses were performed in MATLAB using TopoToolbox (Schwanghart and Scherler, 2014) and the Copernicus digital elevation model (DEM) with 90 m spatial resolution (The European Space Agency, 2021). Mean annual precipitation data for precipitation corrections was obtained at 1 km resolution from the CHELSA global climate model (Karger et al., 2017, 2018). Scientific colormaps used in the figure maps are from Crameri (2023).

### 3.2 Geomorphic Metrics

Cross-divide differences were determined for four metrics--catchment-restricted local relief (CRR), hillslope gradient (HSG), $\chi$, and normalized stream channel steepness ($k_{sn}$) across the entire HDM region. These metrics have been suggested to predict drainage divide migration direction in various tectonic and climatic settings (Willett et al., 2014; Whipple et al., 2017c; Winterberg and Willett, 2019).

Catchment-restricted local relief (CRR) is the maximum elevation difference within a fixed radius window, where the averaging window does not consider points on opposing sides of a specified drainage divide. CRR thus prevents the blurring of relief across drainage divides, which is important when evaluating cross-divide differences. CRR was calculated for the region with a radius of 5 km, similar to the methods of Winterberg & Willett (2019). The stream network was derived and catchment outlets and confluences were defined using TopoToolbox for all streams with a minimum drainage area of 10 km². For every confluence or outlet, the upstream local relief was calculated only within the catchment using the divides to the next catchment as a boundary. Where there is overlap, the catchment with the lower Strahler stream order is used to produce a composite map. The CRR MATLAB script is publicly available on Zenodo (Gelwick and Ott, 2023a). For comparison, traditional local relief (LR) was also calculated with a radius of 5 km (Fig. S2).

Hillslope gradient (HSG) was calculated in degrees using the *gradient8* function in TopoToolbox. This function determines the steepest downward gradient of a DEM pixel using an eight-connected neighborhood.

Stream channel steepness has been shown to be related to erosion rate and has subsequently been widely applied to predict landscape evolution (e.g., Wobus et al., 2006). Combining the stream power incision model with conservation of mass, stream elevation profiles can be described by:

$$dz/dt = U - E = U - KA^mS^n \ , \tag{1}$$

where $U$ is rock uplift rate, $E$ is erosion rate, $K$ is an erodibility constant, $A$ is drainage area, $S$ is channel slope, and $m$ and $n$ are empirical scaling factors (Howard, 1994; Whipple and Tucker, 1999). Over time, given constant $U$ and $K$, a steady-state

landscape is achieved by adjusting channel slope until $E = U$ ($dz/dt = 0$). At steady-state, local channel slope can be expressed as:

$$S = k_s A^{-\theta} , \tag{2}$$

where $k_s$ is channel steepness corrected for drainage area and $\theta$ is stream concavity, which is equivalent to $m/n$ (Hack, 1957; Flint, 1974). Fixing $\theta$ to a reference value ($\theta_{ref}$) enables the comparison of normalized channel steepness ($k_{sn}$) across a given region (Wobus et al., 2006):

$$S = k_{sn} A^{-\theta_{ref}} , \tag{3}$$

$k_{sn}$ can then be used to infer variation in rock uplift rates of steady-state streams, assuming little or no variation in $K$ (similar climate and lithology) in the study region. A best-fit $\theta_{ref}$ of 0.45 was determined for the HDM through Bayesian optimization with the *mnoptim* function in Topotoolbox (Fig. S3). The *mnoptim* function loops through subsets of the drainage network, determines in each iteration the concavity that best linearizes the stream profiles of the subset, and then tests this concavity on the remainder of the network to find the best-fit for the entire drainage network. Given that the strong precipitation gradient in the HDM region could alter the drainage area discharge scaling, we applied a precipitation correction to $k_{sn}$ to get $k_{snP}$ (Ott et al., 2023). Following the methods of Adams et al. (2020), we multiply $A$ by the upstream mean annual precipitation ($P$) for an improved stream discharge proxy:

$$k_{snP} = (AP)^{\theta_{ref}} S , \tag{4}$$

See supplement for results of $k_{sn}$ without the precipitation correction (Fig. S2). For all calculations, a critical drainage area threshold for stream initiation of 5 km$^2$ was used.

Chi ($\chi$) is the horizontal transformation of distance along a stream and was originally developed to reduce noise in stream profiles (Perron and Royden, 2013). Subsequent work argues that the map pattern of $\chi$ can be used to estimate geometric equilibrium or disequilibrium and to infer drainage divide movement in order to achieve geometric equilibrium (Willett et al., 2014). Assuming spatially invariant $U$ and $K$ and integrating Eq. 3, yields:

$$z(x) = z(x_b) + k_{sn} A_0^{-\theta_{ref}} \chi , \tag{5a}$$

and

$$\chi = \int_{x_b}^{x} \left( \frac{A_0}{A(x')} \right)^{\theta} dx' , \tag{5b}$$

where $x_b$ is the stream base-level, and $A_0$ is a reference scaling area (Perron and Royden, 2013). A base-level of 500 m was used for the study area to approximate the elevation at the western edge of the Sichuan Basin. The Sichuan Basin is a part of the stable South China Tectonic Block and serves as a natural base-level for most streams in the HDM via the Yangtze River, which possesses the highest base-level of any major river in the region (Fig. 1a). As with $k_{sn}$, we applied a precipitation correction to $\chi$ to get $\chi_P$ such that:

$$\chi_P = \int_{x_b}^{x} \left( \frac{A_0}{A(x')P} \right)^{\theta} dx' , \tag{6}$$

Results of $\chi$ without the precipitation correction can be found in the supplement (Fig. S2).

## 3.3 Divide asymmetry

Drainage divides for the HDM were determined using the *DIVIDEobj* function in Topotoolbox (Scherler and Schwanghart, 2020). This generates a divide network, similar to a stream network, where divides can be ordered. Divide segments are separated from each other by drainage junctions so that each channel head has a corresponding and unique divide segment. For each divide segment, all pixels draining from the divide to adjacent streams on either side of the divide were used in the calculation for each geomorphic metric (*upslopestats* function). For CRR and HSG, stream pixels were removed, so that only hillslope pixels draining locally into the stream are included. In this way, we ensure that values for divide segments located between interfluves reflect local conditions and not the upstream average of the main channel. The mean metric values for every stream pixel were then projected to the drainage divides (*mapfromnal* function). For $k_{sn}$ and $\chi$, values from the stream were directly projected onto the hillslopes, without averaging, but with prior smoothing of $k_{sn}$ values (*smooth* function).

Divide asymmetry was calculated for each geomorphic metric using a modified version of the *asymmetry* function in Topotoolbox (Scherler and Schwanghart, 2020), where the median of all pixels along the divide was calculated on either side of each divide segment before determining the asymmetry of the segment. This buffers outliers and double-counting of pixels in paired pixel comparisons of the original function. The *asymmetry* function was further modified to ensure that the direction of asymmetry is always perpendicular to the average orientation of the divide segment, which is important for comparison between geomorphic metrics. The magnitude of divide asymmetry was quantified using a modified version of the divide asymmetry index (DAI) proposed by Scherler and Schwanghart (2020):

$$DAI = \left| \frac{\Delta\mu}{\Sigma\mu} \right|, \tag{7}$$

where $\mu$ is the mean value of a given geomorphic metric on either side of a divide segment. By normalizing the across-divide differences by their sum, DAI allows for a simple comparison of asymmetry magnitudes within and across geomorphic metrics. DAI ranges between 0 and 1, for completely symmetric and maximally asymmetric divides, respectively. The MATLAB script we used to calculate DAI for all of the metrics is publicly available on Zenodo (Gelwick and Ott, 2023b). For the purpose of this study, after calculating DAI for each geomorphic metric across the HDM, we restricted our analysis to divides of Strahler order four or greater (Scherler and Schwanghart, 2020). Excluding minor divides reduces noise from small nested catchments and allows us to focus on the signal of large-scale geometric change in the HDM.

## 4 Results

### 4.1 Divide asymmetry by geomorphic metrics

There is strong spatial variation in all measured divide asymmetry metrics across the HDM (Fig. 2). CRR is generally low above 29° N, towards the Tibetan Plateau, and below 26° N (Fig. 2a). Exceptions to this pattern include especially high relief around the Eastern Himalayan syntaxis and in the Longmen Shan region. Between 26° N and 29° N, CRR is generally high, especially in the Three Rivers Area and Daxue Shan. Several patches of low relief occur within this latitude range and

include the Li River catchment, Yanyuan Basin, Anning River catchment, and upper Heng River catchment (numbered 3-6 in Fig. 2d). Above 29° N, a few additional areas with low relief are surrounded by regions of higher relief, including the Yuqu River catchment and Liqiu River catchment (numbered 1 and 2, respectively in Fig. 2d). Variations in HSG closely mirror spatial patterns in CRR, especially in low relief areas (Fig. 2b). Values of $k_{snP}$ are highest in the Three Rivers and around the Eastern Himalayan syntaxis (Fig. 2c). There are stronger contrasts in these areas in the $k_{snP}$ map compared to the $k_{sn}$ map (Fig. S2). Otherwise, variations in $k_{snP}$ closely mimic those described in CRR and HSG. As $\chi_P$ increases with distance from base-level, it is highest in the upper reaches of major rivers in the north (Fig. 2d). However, strong contrasts in $\chi_P$ are observed at catchment boundaries throughout the study region, including around the previously mentioned low-relief features.

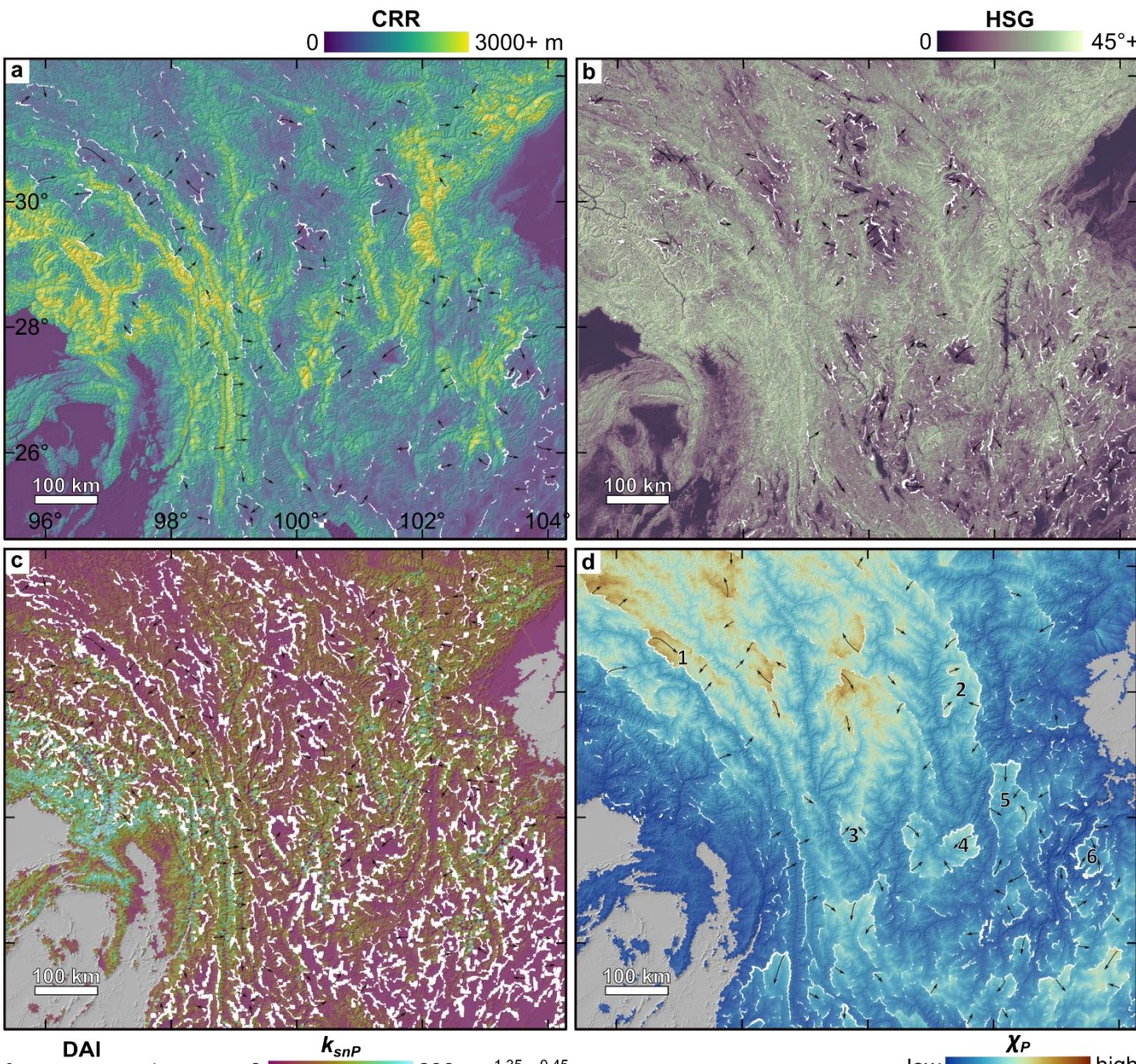

**Figure 2: Geomorphic metrics across the HDM and corresponding divide asymmetry.** Panels show CRR (a), HSG (b), $k_{snP}$ (c), and $\chi_P$ (d). White lines are drainage divides, with thicker lines indicating a higher divide asymmetry index (DAI) for the specific metric. Black arrows show inferred divide migration direction; arrow length does not scale with DAI. Numbers in (d) correspond to low-relief landscape features discussed in the text: (1) Yuqu River catchment, (2) Liqiu River catchment, (3) Li River catchment, (4) Yanyuan Basin, (5) Anning River catchment, and (6) upper Heng River catchment.

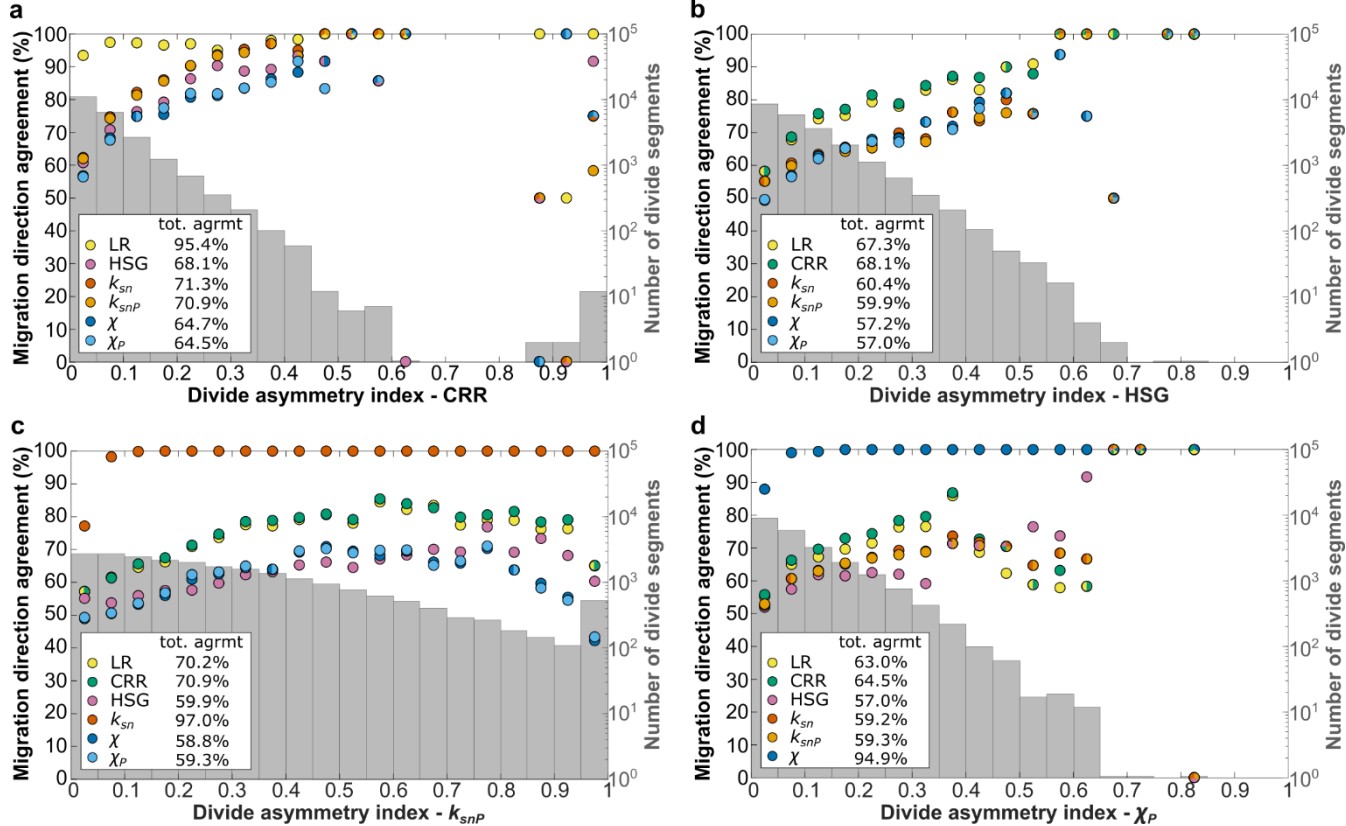

**Figure 3: Plots of percent agreement in divide migration direction between chosen metrics and all calculated metrics (colored points), binned by corresponding divide asymmetry index (DAI) for indicated metric in intervals of 0.05. Grey histograms show the distributions of DAI values in log-scale for each metric. Higher DAI corresponds with increased agreement in migration direction between metrics. Histograms show variability in DAI distributions in different metrics. Plots for LR, $k_{sn}$, and $\chi$ are in Fig. S8.**

As the DAI ranges are distinctly different between geomorphic metrics (Fig. 3, Table S1), we assign a separate threshold for "high," and "low" divide asymmetry in each metric. We define highly asymmetric divides as having DAI values in the 95th percentile for the specific metric. Divide segments with DAI values in the 5th percentile are designated as having low asymmetry. DAI statistics for each metric, including the high and low asymmetry thresholds, can be found in Supplementary Table S1.

Highly asymmetric drainage divides are found across the HDM in all geomorphic metrics (Figs. 4, S4). With the exception of $\chi$ and $\chi_P$, all metrics have a relatively uniform distribution of highly asymmetrical divides across the region. $\chi$ and $\chi_P$ have a higher concentration of highly asymmetric divides to the south and east, possibly because these areas have the lowest elevations in the HDM. $\chi$ and $\chi_P$ have less variation in DAI along continuous divides compared to the other metrics because $\chi$ is integrated at the scale of entire catchments. Drainage divides with low asymmetry are also distributed relatively uniformly in all geomorphic metrics, except for $\chi$ and $\chi_P$, in which fewer are found to the south and east (Figs. 4, S3). Highly asymmetric drainage divides occur in both high- and low-relief areas (Table S2). All geomorphic metrics show high divide asymmetry in the previously identified high- (Eastern Himalayan syntaxis, Three Rivers area, Daxue Shan) and low-relief areas (numbered

in Fig. 4d), except in the Anning River catchment, where only $\chi$ and $\chi_P$ have high asymmetry (Table S2). DAI generally
increases with drainage divide Strahler order (Fig. S5), though this trend is clearest in CRR.

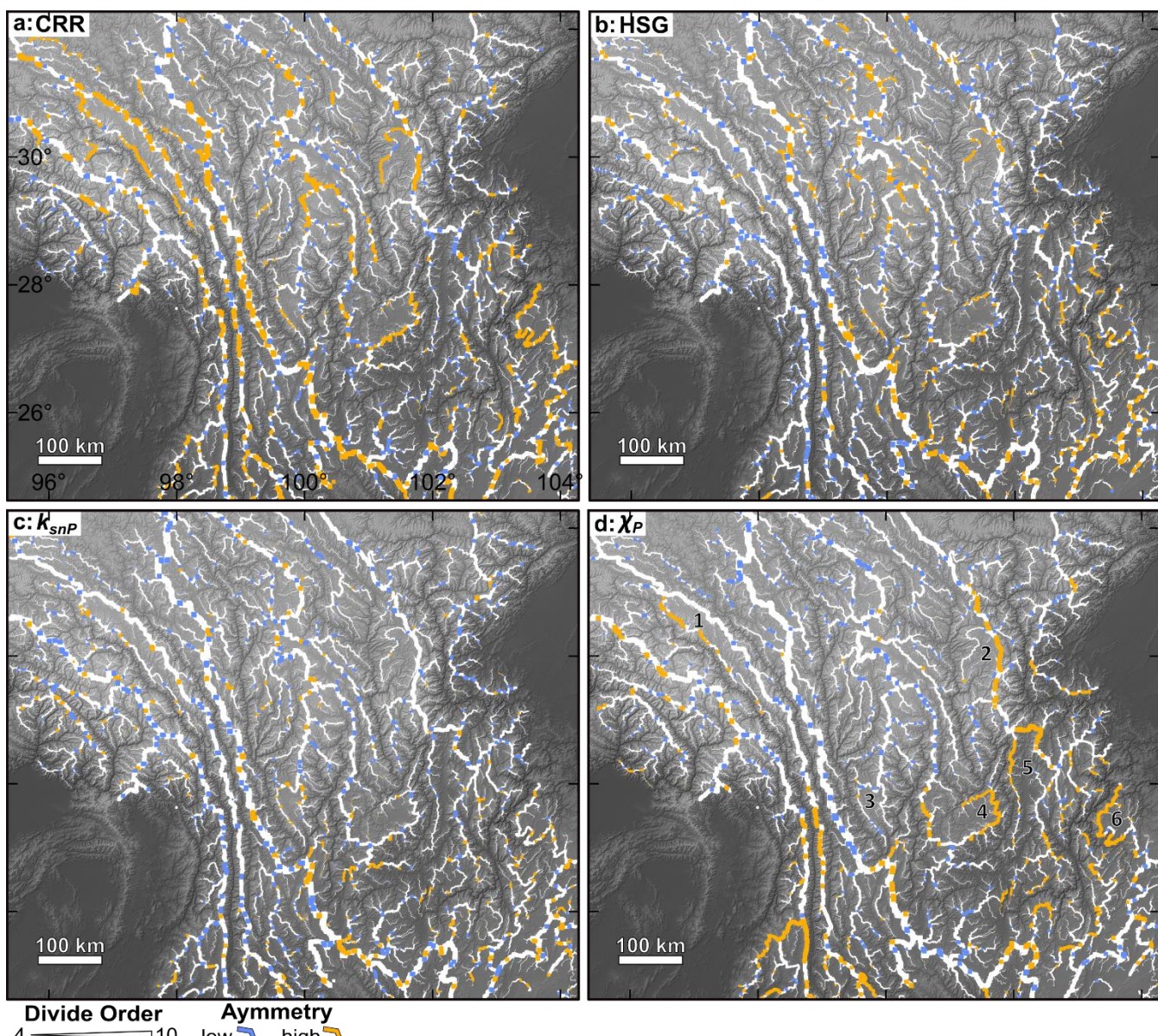

**Figure 4: Locations of drainage divides with high (orange) and low (blue) asymmetry by geomorphic metric, where divide line thickness increases with divide Strahler order (4-10). White divides are not classified as having either high or low asymmetry. Panels include CRR (a), HSG (b), $k_{snP}$ (c) and $\chi_P$ (d). Metric-specific thresholds for high and low DAI can be found in Table S1. Numbers**
**in (d) correspond to low-relief landscape features labelled in Fig. 2d. See Fig. S4 for increased visibility of high and low asymmetry in low order divides.**

Comparing distributions of high and low drainage divides between CRR, HSG, and $k_{snP}$, reveals that 2,915 unique divide segments have high asymmetry and 14,727 unique segments have low asymmetry in at least one metric (Fig. 5). These

groupings overlap in 1,396 divide segments, meaning that 6% of all drainage divides are classed as having both high and low
asymmetry in different metrics. There is no clear spatial pattern of divides with contradicting classifications (Fig. 5). No
drainage divide segments have contradicting asymmetry classifications between closely related metrics (i.e., CRR/LR, $k_{snP}/k_{sn}$,
$\chi_P/\chi$), however some divide segments are classified as having high or low asymmetry in one metric but are unclassified in the
corresponding metric (Fig. S6).

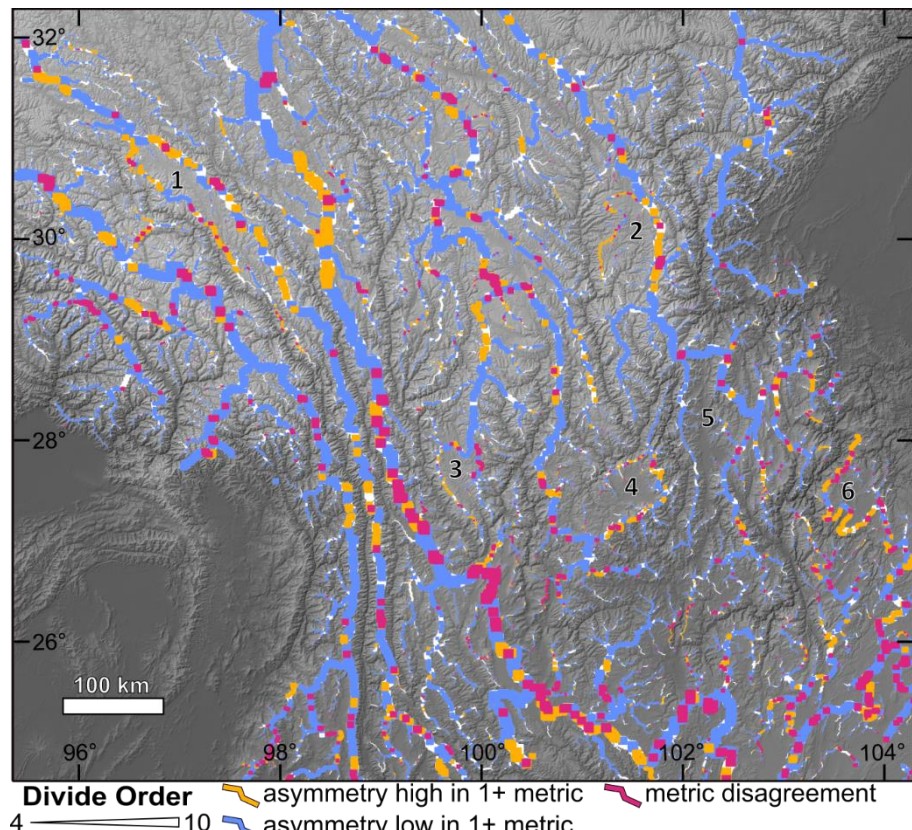

**Figure 5: Spatial comparison of high and low divide asymmetry between local-scale geomorphic metrics, where divide line thickness
increases by divide Strahler order (4-10). Orange divides are those which are classed as highly asymmetric in at least one of the local
geomorphic metrics (CRR, HSG, or $k_{snP}$). $\chi$, $\chi_P$, non-precipitation corrected $k_{sn}$, and local relief are excluded. Blue divides have low
asymmetry in at least one of the local geomorphic metrics. Pink divides have conflicting classifications—they qualify as having high
asymmetry in at least one of the included metrics and low asymmetry in at least one other. Remaining divides are white. Numbers
1-6 correspond to low-relief landscape features labelled in Fig. 2d.**

## 4.2 Agreement in divide asymmetry direction and magnitude between geomorphic metrics

Inferred divide migration directions agree between geomorphic metric pairs for a majority (57% or more) of the 22,837
divide segments analyzed. Closely related metrics, i.e., catchment-restricted and local relief, $k_{sn}$ and $k_{snP}$, $\chi$ and $\chi_P$, agree on
migration direction very strongly with each other with 95.4%, 97.0%, and 94.9% of divide segments agreeing on migration
direction between CRR and local relief, $k_{snP}$ and $k_{sn}$, and $\chi_P$ and $\chi$, respectively (Fig. 3). CRR agrees with other metrics on
divide migration direction more often than any other metric, with agreement ranging between 64.5% ($\chi_P$) and 71.3% ($k_{sn}$) of

divide segments. $\chi_P$ and $\chi$ agree least often with other metrics, the lowest total agreement being with HSG at 57.0% and 57.2% of divide segments for $\chi_P$ and $\chi$, respectively. The geomorphic metrics infer an inward divide migration direction around all of the previously described low-relief landscape features (Fig. 2), except at the headwaters of the Anning River catchment, where only $\chi$ and $\chi_P$ consistently infer an inward migration direction.

Despite overall good agreement in divide migration direction across metrics, the magnitude of divide asymmetry varies significantly by metric. DAI distributions are right-skewed in all metrics, with a majority of divide segments having a low DAI, however, distributions vary (Fig. 3). The metrics $k_{snP}$, $k_{sn}$, and CRR have the broadest DAI distributions (ranging from 0 to 1), however $k_{snP}$ and $k_{sn}$ are less skewed than any of the other metrics (Fig. 3). All other metrics have maximum DAI values of less than 0.9, with fewer than 20 divide segments exceeding a DAI of 0.7 in any single metric.

Divide segments with higher magnitudes of asymmetry are more likely to agree on migration direction across multiple metrics. Figure 3 shows a strong positive correlation between increasing DAI and percent agreement on divide migration direction in all metrics. For instance, CRR shows more than 75% agreement in migration direction for all metrics if the DAI is greater than 0.2. However, this relationship becomes noisy at high DAI values for which most metrics lack a sufficient sample size.

Except between closely related metrics (i.e., CRR/LR, $k_{snP}/k_{sn}$, $\chi_P/\chi$), the Pearson correlation coefficient of DAI values is ≤0.30 for all metric comparisons (Fig. 6). Consistent with divide migration direction agreement, the highest Pearson correlation coefficients correspond with CRR, with values ranging between 0.21 to 0.30 (Fig. 6). Despite a high Pearson correlation coefficient between CRR and LR of 0.63, correlation coefficient values for local relief with other metrics are consistently the lowest with values not exceeding 0.19 (Fig. 6).

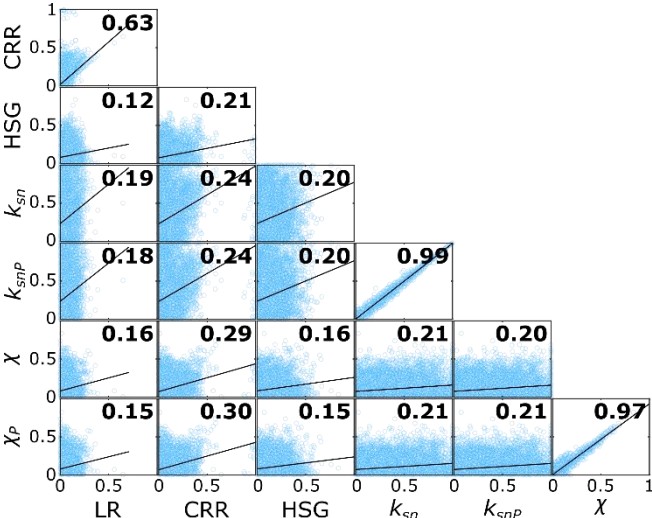

**Figure 6: Pearson's coefficient of variation of divide asymmetry index (DAI) between each pair of metrics (bold, black values). Blue circles show distributions of DAI for each comparison with linear fit indicated by black lines.**

**4.3 Comparison between integral and local-scale geomorphic metrics**

There is a fundamental difference between the local-scale metrics of divide asymmetry (LR, CRR, HSG, $k_{sn}$, $k_{snP}$) and the integral quantities ($\chi$, $\chi_P$) that complicates the interpretation of disagreement between them. The integral quantities are measures of geometric disequilibrium within a catchment but do not necessarily relate directly to the instantaneous or current asymmetry of a specific divide segment. Furthermore, they can be affected by spatial variations in uplift rate or channel bed erodibility occurring anywhere downstream of the assessment point. The integral metrics can thus reflect transient processes

in distal parts of the catchment, rather than processes local to a divide (Whipple et al., 2017c; Forte and Whipple, 2018). Agreement between the local and integral quantities indicates that spatial variations in uplift rate or erodibility are unlikely and that internal transients, due to processes such as river capture have propagated upstream to reach the divides resulting in a quasi-stable state (Beeson et al., 2017). $\chi_P$ agrees with a majority (2+) of the other geomorphic metrics for 69.2% of the 22,837 divide segments analyzed in the HDM. As shown in Figure 3, $\chi_P$ is more likely to predict a contrary divide migration direction

when it, or the metric it is being compared to, has a low DAI. When only divide segments for which all four metrics, including $\chi_P$, have a minimum DAI of 0.05 are counted (n=3,068), $\chi_P$ agrees with at least two of the three other geomorphic metrics 87.2% of the time. At a DAI threshold of 0.1 (n=537), the agreement increases to 92.9%. Disagreements between $\chi_P$ and the other metrics are found across the entire HDM region, but certain regions have higher concentrations of disagreement (e.g., Eastern Himalayan syntaxis, boundary between Dadu River and Anning River catchments, southwest and southeast margins

of study area; Fig. 7).

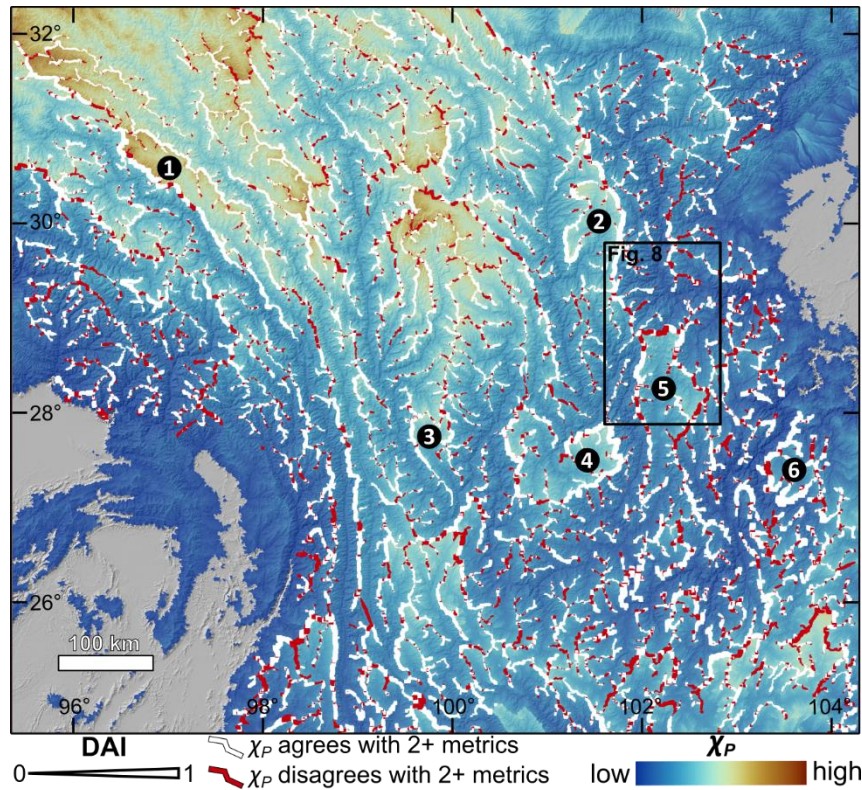

**Figure 7: Map of $\chi_P$ (background) with lines representing drainage divides. Divides are colored by agreement/disagreement in migration direction between $\chi_P$ and a majority of selected geomorphic metrics (i.e., CRR, HSG, $k_{snP}$). White divides indicate agreement between $\chi_P$ and at 2+ other metrics. Red divides indicate disagreement between $\chi_P$ and 2+ other metrics. Divide line thickness corresponds to DAI of $\chi_P$, with thicker divides having greater asymmetry. Black numbered circles indicate low-relief catchments identified as undergoing drainage area-loss feedback: (1) Yuqu River catchment, (2) Liqiu River catchment, (3) Li River catchment, (4) Yanyuan Basin, (5) Anning River catchment, and (6) upper Heng River catchment.**

## 5 Discussion

### 5.1 Comparison of geomorphic metrics and implications in the HDM

Overall, geomorphic metrics agree on divide migration direction in a majority of cases (Figs. 3 and 7), with total

agreement between any two metrics ranging from 57% to 97%. Where geomorphic metrics contradict each other, divide

asymmetry is typically low, indicating a more stable divide (Fig. 3). The variation in DAI across metrics for any given divide

segment is the result of the relative spatial scales and sensitivity of each metric. Metrics are computed on different spatial and,

consequently, temporal scales; for example, $\chi_P$ is integrated across entire catchments and represents long-term trends in

landscape evolution (Beeson et al., 2017; Whipple et al., 2017c; Forte and Whipple, 2018; Scheingross et al., 2020), while

HSG is averaged across individual divide segments and adjusts to local, short-term changes to catchment structure.

Additionally, it should be noted that some metrics may have distinct threshold behavior. For instance, it has been argued that HSG and relief reach a threshold above which they do not correlate with erosion rate (Burbank et al., 1996; Montgomery, 2001), especially in tectonically active areas like the HDM. The histogram of HSG values in the HDM (Fig. S7) shows a marked decline in frequency above a slope of 30° in the HDM, indicating that hillslopes are reaching a threshold. This supports the $30 \pm 5°$ HSG threshold determined by Liu et al. (2021b) based on the mode slope values of major catchments in the HDM. Channel steepness ($k_{sn}$) also exhibits non-linear and potentially threshold behaviour (Hilley et al., 2019). This is consistent with the metric-dependent ranges and distributions of DAI (Table S1, Fig. 3) and may also explain why DAI and divide Strahler order have a stronger positive relationship in CRR than in the other metrics (Fig. S5). Therefore, identical DAI values in different metrics would likely be associated with different divide migration rates. Due to this potential for threshold behavior and the expectation that a given DAI value corresponds to a different erosion rate in each metric, we use metric-specific thresholds to distinguish drainage divides with high and low asymmetry. We consider drainage divides in the 95th and 5th percentiles of each geomorphic metric to have high and low asymmetry, respectively (Table S1, Fig. 3).

Contradicting predictions for divide migration direction between metrics may be the result of local-scale metrics (i.e., CRR, HSG, $k_{snP}$) responding to variations in rock uplift and erodibility. This is supported by high concentrations of disagreement between $\chi_P$ and local-scale metrics in areas with known localized uplift (e.g., Eastern Himalayan Syntaxis, Longmen Shan; Fig. 7). However, divide migration direction inferred by $\chi_P$ still agrees with at least two of the three local metrics 69.2% of the time, especially in divides with strong $\chi_P$ asymmetry (83.7%). This suggests that, despite several areas of localized uplift and heterogeneous lithology across the HDM (Fox et al., 2020; Fig. S1), changes in uplift and erodibility are small enough, or slow enough, that $\chi_P$ values at the divides remain dominated by the length of the rivers and their area distributions in most places.

Other processes can further complicate the divide asymmetry relationship between geomorphic metrics with different spatial and temporal scales. For example, glaciers, which are concentrated along drainage divides, could have altered catchment relief and slope through planation, over-deepening, and/or asymmetrical erosion (Lai and Huppert, 2023). Interestingly, however, drainage divides of the low-relief Yiqui River catchment (labelled "2" in Fig. 7) exhibit good agreement between $\chi_P$ and local metrics, despite the area's glacial history (Zhang et al., 2016). Additional reported drivers of cross-divide erosional differences include river capture (Willett et al., 2014; Beeson et al., 2017; Scheingross et al., 2020), non-uniform bedrock erodibility (Gallen, 2018; Wang et al., 2023; Mitchell and Forte, 2023), tectonic advection (Chen et al., 2021; He et al., 2021; Mitchell and Forte, 2023), changes in precipitation patterns (Bian et al., 2024), landsliding (Dahlquist et al., 2018), endorheic lake expansion (Liu et al., 2021a), and autogenic fluvial processes (Scheingross et al., 2020). The discordance in divide migration direction between $\chi_P$ and local metrics in weakly asymmetric divides suggests that migration direction is more susceptible to minor shifts in erosion rates on one side of the divide, such as those caused by lithologic differences or autogenic fluvial processes, when the drainage network is geometrically balanced.

Our results highlight the importance of the concept of quasi-topographic equilibrium when interpreting divide migration metrics, in which landscape transience can be characterized by a combination of long-term adjustments towards a

quasi-equilibrium state and short-term perturbations driven by local geometry and transient mechanisms (Beeson et al., 2017). Although we find evidence for local tectonic and geomorphic perturbations driving local divide migration, the drainage network is dominated by quasi-equilibrium conditions in which most river profile perturbations have reached the water divide and are driving steady migration of the divides and asymmetry in all indicators.

## 5.2 Continental-scale landscape transience in the HDM

Except in the case of $\chi_P$, there is no clear spatial pattern in DAI magnitudes in the HDM, with high and low values distributed relatively uniformly (Fig. 4). The greater incidence of divides with high $\chi_P$ asymmetry in the east and south may be due to these areas having lower elevations. The existence of strongly asymmetrical drainage divides throughout the HDM combined with the lack of spatial trends is evidence of landscape transience at a continental scale.

While DAI has not been calibrated to absolute rates of drainage divide migration for any metric, divides with strong asymmetries may migrate relatively quickly. For instance, divides with a $k_{snP}$ DAI of 0.564, our threshold for high $k_{snP}$ asymmetry, would have a 56.4% difference in cross-divide erosion rates based on the Stream Power Law (Eq. 1), assuming a linear relationship between channel steepness and erosion rates (where $n = 1$).

Catchments surrounded by highly asymmetric divides migrating inward are particularly suggestive of active drainage reorganization, with the bounded catchments losing drainage area to their neighbors (Whipple et al., 2017a, b; Willett, 2017; Fox et al., 2020). This pattern was identified and interpreted by Yang et al. (2015) as representative of catchments which have lost headwaters to a large river capture event and are subsequently collapsing due to their reduced erosional power, in what has been referred to as an area-loss feedback (Willett et al., 2014). Many examples in the region, including the Yuqu River catchment, Liqiu River catchment, Li River catchment, Yanyuan Basin, and upper Heng River catchment (numbered in Figs. 2d, 3d, 5 and 7), fit this model well.

However, in areas in the HDM without any clear captures or where local-scale metrics contradict $\chi_P$ at catchment boundaries, the source of transience is unclear. In these cases, a local analysis of landscape features is required and additional drivers, such as those discussed in the previous section, should be considered. Below, we include one such analysis from the Anning River catchment in which we examine how discrete river capture can temporarily alter drainage divide asymmetry.

## 5.3 Influence of discrete river capture on short-term divide asymmetry

Discrete river captures can temporarily alter divide asymmetry relationships while the stream channels and corresponding hillslopes adjust to the change in erosive power (Prince et al., 2010, 2011; Willett et al., 2014; Whipple et al., 2017c). Notably, $\chi_P$ predicts a migration direction contradictory to the other metrics across much of the drainage boundary between the Dadu and Anning rivers (Figs. 7, 8), where a major river capture recently occurred and ongoing divide migration is well-documented (Yang et al., 2020). We expect that the discrepancy is a direct consequence of the planform adjustments occurring to the river network following capture of the former upper Anning to the modern Dadu ~2 Ma (Yang et al., 2020). The modern upper Dadu is suggested to have been captured at the location of the "initial windgap" circled in Figure 8a. The

capture event resulted in the formation of a new drainage divide just south of the capture point that has since migrated to the south by approximately 40 km. This divide migration has reversed the river along the former path of the Anning and led to the capture of the adjacent tributaries in a pattern characteristic of headward extension of the Dadu tributary (Bishop, 1995; Harel et al., 2022; Fig. 8a). A consequence of this process is that much of the modern drainage divide between the Anning and the Dadu formed as a divide fully internal to the Anning catchment and has only recently become the main divide between the Anning and the Dadu through capture of a lateral tributary (Fig. 9a, "pre-capture divide").

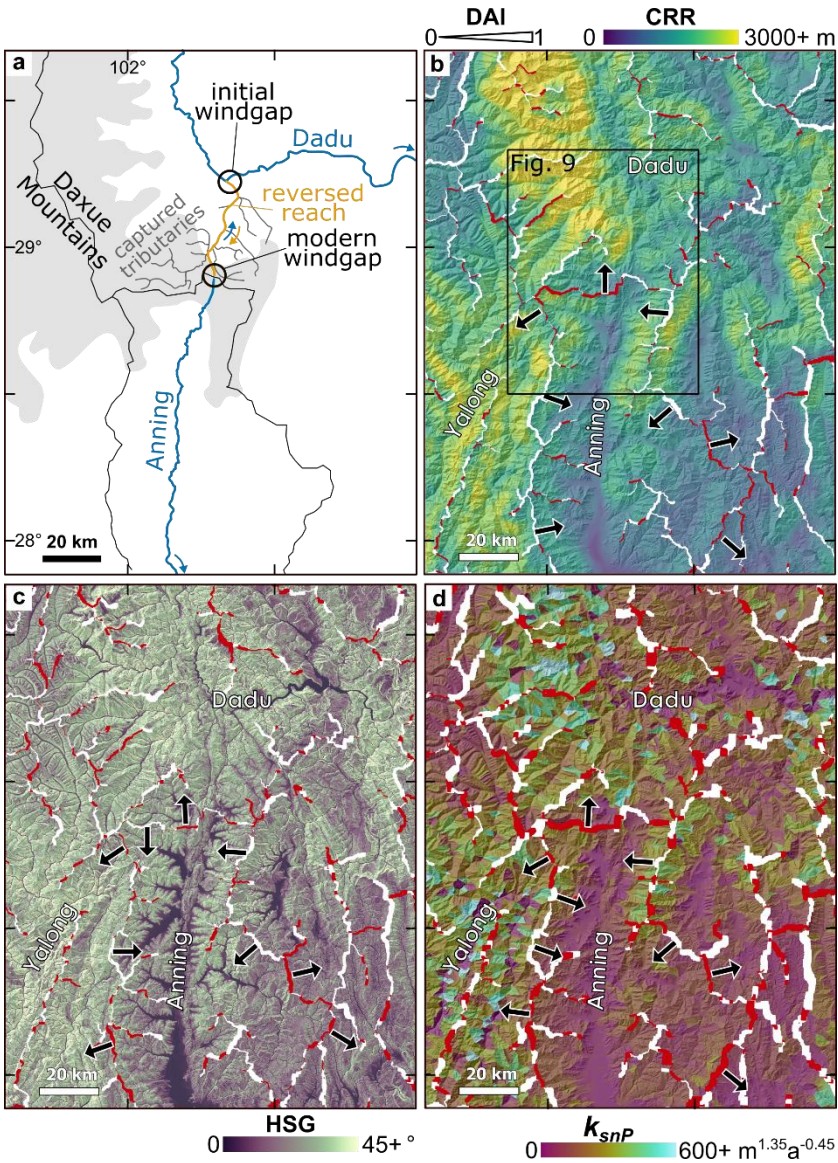

**Figure 8: (a) Schematic of drainage capture between the modern Dadu and Paleo-Dadu-Anning rivers. Blue lines show current flow paths of Dadu and Anning rivers; orange line shows the river reach that has been reversed since the initial capture; dark grey lines**

are former tributaries to the Anning that have been captured by the Dadu; the main drainage divides are outlined in black. Blue arrows show modern flow direction, orange arrow shows pre-capture flow direction. Initial capture point (initial windgap) and modern windgap are indicated by black circles. Grey shading shows approximate modern extent of the Daxue Mountains. (b) Map of CRR and drainage divides (white and red lines). Red divides indicate disagreement between $\chi_P$ and 2+ other metrics. Divide line thickness corresponds to DAI of CRR, with thicker divides having greater asymmetry. Black arrows show inferred divide migration direction based on CRR. (c) Map of HSG and drainage divides (white and red lines). Red divides indicate disagreement between $\chi_P$ and 2+ other metrics. Divide line thickness corresponds to DAI of HSG. Black arrows show inferred divide migration direction based on HSG. (d) Map of $k_{snP}$ and drainage divides (white and red lines). Red divides indicate disagreement between $\chi_P$ and 2+ other metrics. Divide line thickness corresponds to DAI of $k_{snP}$. Black arrows show inferred divide migration direction based on $k_{snP}$.

To support this model of headward expansion of the Dadu, we examine the stream profiles of opposing side tributaries from this segment of the main Dadu-Anning drainage divide, starting from the approximate elevation of the Anning River outlet at 1,000 m (Fig. 9). A clear change in channel steepness is observed in the Dadu stream profile, with a low-gradient upstream segment that begins above a dam that was constructed ca. 2013 and likely corresponds with a natural knickpoint (Fig. 9b). The knickpoint also coincides with where disagreement between $\chi_P$ and local-scale metrics begins along the Dadu-Anning drainage divide (Fig. 9a). No significant change in channel steepness is observed in the Anning profile, despite also crossing a dammed section (Fig. 9b). Notably, the flowpaths of the opposing tributaries' main trunks coincide with, and are likely controlled by, a branch of the Anninghe Fault.

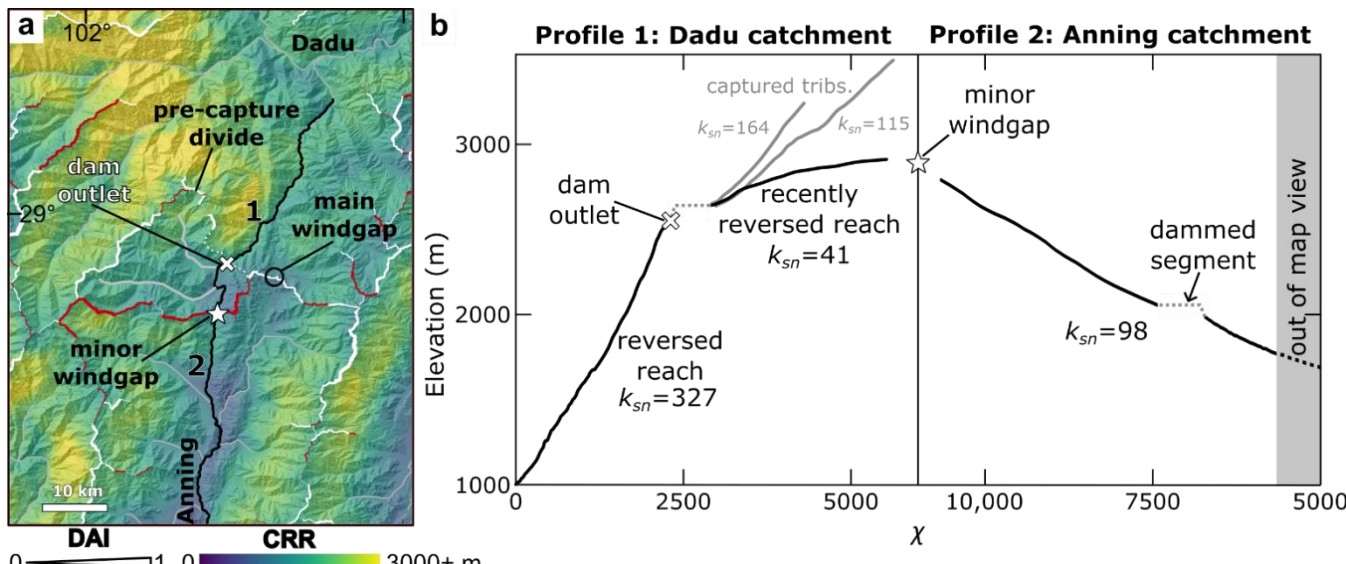

Figure 9: (a) CRR map from Fig. 8b. Black lines highlight section of the Dadu (1) and Anning (2) tributaries shown as opposing stream profiles in b. Remaining stream network is shown in grey. White "X" marks the knickpoint below the portion of the Dadu tributary recently acquired through lateral river capture. Drainage divides are shown in red ($\chi_P$ disagrees with CRR) and white ($\chi_P$ agrees with CRR). Divide line thickness corresponds to DAI of CRR. Location of the main, modern windgap between the Dadu and Anning catchments (same as in Fig. 8a) is indicated by the black circle. The "pre-lateral capture divide" was the drainage boundary between the Dadu and Anning catchments before the recent lateral river capture. White dotted line outlines the portion of the pre-capture divide not included in the analysis due to low Strahler order. Note that CRR is markedly lower on what was the Anning side of the pre-capture divide and predicts the same migration direction as $\chi_P$. (b) Opposing stream profiles ($\chi$ vs. elevation) corresponding to segments of Dadu and Anning rivers shown in a, initiating from the white star labeled "minor windgap." For plotting purposes, $\chi$ baselevel is set at 1000 m (approx. elevation of Anning outlet). Abrupt reduction in slope (mean $k_{sn}$ of 41 vs. 327) above the knickpoint (white "X") in the Dadu profile corresponds with the recent lateral river capture from the Anning tributary

**shown in profile 2. Grey profiles are side branches of the recently captured tributary which have not been reversed. Prior to capture, the stream segments above the knickpoint flowed into the Anning.**

The Dadu knickpoint is concurrent with the lateral tributary's outlet, suggesting that the low-gradient tributary was captured by the Dadu, having formerly flowed into the Anning via the corresponding windgap (Fig. 9a, "minor windgap"). The initial capture of the tributary led to a flow reversal along its main trunk (black line upstream of dam outlet in Fig. 9) and the capture of additional tributary branches (solid grey lines upstream of dam outlet in Fig. 9). River capture triggers a kinematic wave with a response time determined by the erosive power of the expanded catchment (Weissel and Seidl, 1998; Seidl et al., 1992; Whipple and Tucker, 1999). Immediately following a river capture event, the diverted stream segment and

corresponding hillslopes will not yet have adjusted to the new base-level. Local-scale geomorphic metrics will predict a false migration direction until the kinematic wave has propagated through the entire captured reach. Thus, the apparent disagreement between $\chi_P$ and local-scale metrics at the Dadu-Anning divide should only persist for a short time, while $\chi_P$ continues to predict the correct long-term migration direction. This interpretation is supported by the agreement between $\chi_P$ and local-scale metrics on divide migration direction in what would have been the Dadu-Anning drainage divide pre-capture (Fig. 9a).

**6 Conclusions**

We investigated drainage divide asymmetry using four geomorphic steepness or elevation metrics (CRR, HSG, $k_{snP}$, and $\chi_P$) to understand the spatial and temporal patterns of geometric transience in the HDM river network. We find clear evidence of widespread transience through a high incidence of strongly asymmetrical divides throughout the HDM in all geomorphic metrics.

Our findings demonstrate that $\chi_P$, which is a proxy for the long-term landscape evolution, generally agrees with local-scale geomorphic metrics that are only sensitive to local erosional processes, especially in areas of pronounced asymmetry. This suggests that most of the landscape is in a quasi-equilibrium state where short-term and long-term disequilibrium indicate the same sense of geometric change in the drainage network. It also suggests that variations in climate, lithology, and uplift rate are small compared to the normalized length of rivers. However, disruptions to long-term trends in landscape evolution,

as indicated by discrepancies between $\chi_P$ and local metrics, can arise from localized tectonic or geomorphic events. This is exemplified by the Dadu-Anning drainage divide scenario where $\chi_P$'s asymmetry contrasts with the other metrics, despite accurately forecasting the Anning River's ongoing capture by the Dadu River.

In addition to delineating geometrically transient and stable areas in the HDM landscape at high resolution, our work demonstrates the important differences in local and regional predictors of transience. We show how disagreement across

timescales can be used as evidence for discrete transient events, such as river capture, whereas consensus amongst all indicators is evidence for a state of quasi-equilibrium in which geometric change occurs in the drainage network in a near-steady fashion. These insights are broadly applicable to the study of landscape transience and drainage divide asymmetry in other regions, regardless of their tectonic, geologic, or climatic setting.

## Code and data availability

All MATLAB scripts used in this study are available online at: https://doi.org/10.5281/zenodo.8363610 and https://doi.org/10.5281/zenodo.8416263. All other necessary information and data for study replication are provided in the main text and the accompanying Supplement.

## Author contribution

KG: conceptualization, methodology, formal analysis, investigation, software, writing – original draft preparation; SW: supervision, conceptualization, funding acquisition, writing – review and editing; RY: conceptualization, funding acquisition, writing – review and editing.

## Competing interests

The authors declare that they have no conflict of interest.

## Acknowledgements

This research was supported by the Swiss National Science Foundation (grant no. 189846) and the National Natural Science Foundation of China (grant no. 41961134031). This manuscript benefited from conversations with Richard F. Ott, Helen W. Dow, and Kimberly L. Huppert. We also thank Ott for his MATLAB assistance. Additionally, we express our sincere gratitude to the two anonymous referees and the associate editor, Fiona Clubb, for their thoughtful and constructive comments.

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
