# Peer review of "Geomorphic indicators of continental-scale landscape transience in the Hengduan Mountains, SE Tibet, China"

_EGUsphere, 2023_

## Author Comment (AC1)

**Response to Anonymous Referee #1**

*"In this contribution, Gelwick et al., presents an analysis of topography mostly associated with drainage divides in the Hengduan Mountains, with an additional focus on comparing the implications and predictions of a variety of divide stability / mobility metrics. Overall, the paper is well organized and clearly written. Drainage divide stability remains a topic of general interest within the geomorphology community, especially so in this particular region of the world, so the paper seems appropriate for ESurf in terms of audience. The majority of my comments on the paper are minor, which I can classify into three broad themes that I summarize below and then flesh out in line-by-line comments.*

*There are a variety of places where it seems like it would be good to cite additional papers and/or acknowledge prior work more clearly.*

*In part related to the first point, in a few places in the manuscript the authors seem to imply that the comparison between divide metrics is novel and/or that the general conclusions of comparing different divide metrics, either in the abstract or in specific landscapes is new, when in fact there are a variety of efforts in the prior literature, some that they cite and others that they don't (e.g., Forte & Whipple, 2018; Sassolas-Serrayet et al., 2019; Ye et al., 2022; Zhou et al., 2022). There is still definitely value in the detailed comparisons presented here, but at the same time, it would be good to acknowledge that many of these same points have been demonstrated by others before."*

We agree with the reviewer that additional references are justified and have modified several sentences to include all of the suggested references and a few others from the literature. In addition to those added in response to other comments and described later, we modified the following passages (blue text is new):

L56-58: "Common metrics include mean hillslope gradient, mean local relief, stream channel steepness, channel head elevation, and hillslope curvature measured near the divide (Hurst et al., 2013; Whipple et al., 2017c; Forte and Whipple, 2018; Scherler and Schwanghart, 2020; Zhou et al., 2022)."

L59-62: "In addition to these local-scale metrics, $\chi$, a transformed variable of the along-stream distance (Perron and Royden, 2013), has been widely applied to assess the general geometric stability of the drainage network pattern with the assumption that planform patterns in $\chi$ should be reflected in the distribution of divide elevation and symmetry (Willett et al., 2014; Beeson et al., 2017; Sassolas-Serrayet et al., 2019; Ye et al., 2022; Zhou et al., 2022)."

L69-70: "Metrics which reflect local erosion and uplift dynamics are thus more reliable predictors of instantaneous motion of specific drainage divides at a specific time (Whipple et al., 2017c; Sassolas-Serrayet et al., 2019; Dal Pai et al., 2023)."

L74-77: "Local-scale metrics are also subject to variations in local physical properties and transients and regularly exhibit large variability along drainage divides, as well as internal contradictions between metrics (Sassolas-Serrayet et al., 2019; Dal Pai et al., 2023). To mitigate this, studies often combine multiple metrics and/or take the mean value of all catchments along each side of a main drainage divide (Forte and Whipple, 2018; Sassolas-Serrayet et al., 2019; Zhou et al., 2022; Dal Pai et al., 2023)."

L315-318: "Metrics are computed on different spatial and, consequently, temporal scales; for example, $\chi_P$ is integrated across entire catchments and represents long-term trends in landscape evolution (Beeson et al., 2017; Whipple et al., 2017c; Forte and Whipple, 2018; Scheingross et al., 2020), while HSG is averaged across individual divide segments and adjusts to local, short-term changes to catchment structure."

L331-333: "This suggests that, despite several areas of localized uplift and heterogeneous lithology across the HDM (Fox et al., 2020), changes in uplift and erodibility are small enough, or slow enough, that $\chi_P$ values at the divides remain dominated by the length of the rivers and their area distributions in most places."

L338-342: "Additional reported drivers of cross-divide erosional differences include river capture (Willett et al., 2014; Beeson et al., 2017; Scheingross et al., 2020), non-uniform bedrock erodibility (Gallen, 2018; Wang et al., 2023; Mitchell and Forte, 2023), tectonic advection (Chen et al., 2021; He et al., 2021; Mitchell and Forte, 2023), landsliding (Dahlquist et al., 2018), endorheic lake expansion (Liu et al., 2021a), changes in precipitation patterns (Bian et al., 2024), and autogenic fluvial processes (Scheingross et al., 2020)."

L360-361: "Catchments surrounded by highly asymmetric divides migrating inward are particularly suggestive of active drainage reorganization, with the bounded catchments losing drainage area to their neighbors (Whipple et al., 2017a, b; Willett, 2017; Fox et al., 2020)."
* * *
*"Finally, there could be some additional discussion of the methods in terms of how the values of the metrics are considered with respect to each other. At present, the methods rely heavily on readers knowing the specific operation of the referenced TopoToolbox functions to sort of follow what is being done, in even in the event that you do, it remains unclear exactly how they're treating some of the values. I highlight a specific example in the line-by-line comments."*

We agree with the reviewer and, as discussed in a later response, have revised the final manuscript to include more detailed methods.
* * *
*"Line-by-line comments:*

*L44-51: In this section, it seems worthwhile to highlight that the interpretation of this landscape in the context of surface uplift from drainage capture is not without controversy (Whipple, DiBiase, et al., 2017a, 2017b; Willett, 2017)."*

Line 47 of the original submission was meant to indicate that the drainage-area exchange hypothesis is not without controversy: "Alternative explanations for these low-relief features include a delayed incisional response in small tributaries to propagating tectonic uplift from the ongoing India-Eurasia collision (Clark et al., 2006) and/or glacial planation (Zhang et al., 2016)."

To further balance the presentation of previous interpretations we add additional references (including those suggested by the reviewer) and expand the original paragraph discussing low-relief features in the HDM as follows, where new text is blue:

"A prime example of this connection between geometric network change and topography is observed in the high-elevation, low-relief areas scattered throughout the Hengduan Mountains (HDM), Southeast Tibet. Many of these low-relief features have been interpreted to result from river capture, where drainage area loss inhibits the ability of catchment erosion to keep pace with background (Yang et al., 2015; Willett, 2017; Fox et al., 2020; Yuan et al., 2021). This hypothesis is supported by several major river captures in the HDM which indicate significant, ongoing drainage reorganization in the region (Clark et al., 2004; Zheng et al., 2021). While few of these captures have been decisively dated, several have been confirmed or estimated to have occurred in the last 2-4 Ma (Kong et al., 2012; Liu et al., 2020; Sun et al., 2020; Yang et al., 2020). Another explanation for these low-relief features is a delayed incisional response in small tributaries to propagating tectonic uplift from the

ongoing India-Eurasia collision (Clark et al., 2006; Whipple et al., 2017a, b). Glacial planation may have also played a role in their formation in previously glaciated areas (Zhang et al., 2016). Identifying transients in the river network can help to diagnose the origins of low-relief features in the HDM and distinguish between these hypotheses (Whipple et al., 2017a, b; Willett, 2017; Fox et al., 2020). Despite its critical role in shaping the landscape, the prevalence, intensity, and spatial distribution of geometric transience has not been systematically measured across the HDM on a large scale.
* * *
*"L58: It might be prudent to add Forte & Whipple, (2018) to this list as the use of some of the metrics you list were more formally defined there as opposed to the cited Whipple et al., (2017)."*

We agree with the reviewer and have added Forte & Whipple (2018) to the cited references.
* * *
*"L180-182: A minor quibble, but while it's clear that you're calculating the same thing as Adams et al., (2020), is there a demonstrable reason why you're not using the same name as in Adams or other subsequent papers (e.g., Leonard et al., 2023; Leonard & Whipple, 2021)? While I would tend to agree that ksnP might be a more apt name since it incorporates a routed version of mean annual precipitation and thus is not truly discharge (as is effectively implied by calling it ksnQ as in Adams, etc.), I also would argue that it's generally a bad practice to knowingly introduce ambiguity into the literature by arbitrarily renaming a quantity that has been given a particular name in multiple publications."*

We understand the reviewer's concern and gave thought to the terminology. However, we decided to follow the recently published paper by Ott et al. (2023) and use the term $k_{snP}$. As mentioned by the reviewer, the reason is that it is a more direct representation of the data going into the calculation. We think that the term $k_{snQ}$ should be reserved for instances where actual discharge estimates (e.g., stream gage records, satellite-derived $P$ – evapotranspiration data) are being used. While we might be furthering the ambiguity introduced by Ott et al. (2023), we believe that in the long-term it is better to advertise the use of the more accurate term in the geomorphic community. We have added a reference to Ott et al. (2023) when $k_{snP}$ is introduced in the Methods section to reduce confusion.
* * *
*"L201-205: This could be explained a little better. If I follow what you're doing, you calculate the mean upstream value of a given metric for the entire drainage network and then map values from the streams onto divides, which effectively "follows" the FLOWobj up the stream to divide segments? If that is correct, it seems like there should be a little more discussion of the implications of some of these. For example, in a case where a divide is basically between interfluves, would the upstream mean of the main channels (that are nominally orthogonal to this portion of the divide) be mapped with values from these main channels? If that's the case, is the across divide contrast relevant? It's easier to think about a scenario where a divide is between two channel heads with accumulating area above them, but in this case, it's not necessarily clear whether this method is appropriate for all metrics. Specifically, if you're treating ksn / ksnP in this way, that seems problematic as the upslope mean of ksn above a channel head would be basically the colluvial portion of the profile (where ksn is probably not really a valid metric to calculate). Clarification on these points would help readers understand both what you're doing, but also how to interpret your results."*

We thank the reviewer for this comment and agree that the explanation in the original manuscript was vague. For improved clarity, we have expanded the paragraph spanning lines 199-214 in the Methods section as follows (blue text is new):
"Drainage divides for the HDM were determined using the *DIVIDEobj* function in Topotoolbox

(Scherler and Schwanghart, 2020). This generates a divide network, similar to a stream network, where divides can be ordered. Divide segments are separated from each other by drainage junctions so that each channel head has a corresponding and unique divide segment. For each divide segment, all pixels draining from the divide to adjacent streams on either side of the divide were used in the calculation for each geomorphic metric (*upslopestats* function). For CRR and HSG, stream pixels were removed, so that only hillslope pixels draining locally into the stream are included. In this way, we ensure that values for divide segments located between interfluves reflect local conditions and not the upstream average of the main channel. The mean metric values for every stream pixel were then projected to the drainage divides (*mapfromnal* function). For $k_{sn}$ and $\chi$, values from the stream were directly projected onto the hillslopes, without averaging, but with prior smoothing of $k_{sn}$ values (*smooth* function).

Divide asymmetry was calculated for each geomorphic metric using a modified version of the *asymmetry* function in Topotoolbox (Scherler and Schwanghart, 2020), where the median of all pixels along the divide was calculated on either side of each divide segment before determining the asymmetry of the segment. This buffers outliers and double-counting of pixels in paired pixel comparisons of the original function. The *asymmetry* function was further modified to ensure that the direction of asymmetry is always perpendicular to the average orientation of the divide segment, which is important for comparison between geomorphic metrics. The magnitude of divide asymmetry was quantified using a modified version of the divide asymmetry index (DAI) proposed by Scherler and Schwanghart (2020):

$$DAI = \left| \frac{\Delta\mu}{\sum \mu} \right|,$$
(7)

where $\mu$ is the mean value of a given geomorphic metric on either side of a divide segment. By normalizing the across-divide differences by their sum, DAI allows for a simple comparison of asymmetry magnitudes within and across geomorphic metrics. DAI ranges between 0 and 1, for completely symmetric and maximally asymmetric divides, respectively. The MATLAB script we used to calculate DAI for all of the metrics is publicly available on Zenodo: https://doi.org/10.5281/zenodo.8416264."

For easy reference, we have also added direct links to the relevant code in the Methods section.
* * *
*"L243-255: Throughout this section, you refer to supplemental figures S2 and S3 a lot, making it pretty hard to follow this section without referring to the supplement many times. I wonder if it might be better to move these two figures to the main text since you rely on them heavily."*

This is a valid point. In the revised manuscript, we move Fig. S3 to the main text (now Fig. 4). We chose to leave Fig. S2 in the supplement, but have added a simpler version to the main text that includes the "main" metrics (CRR, HSG, $k_{snP}$, and $\chi_P$). This new figure is shown below:

[Figure]

**Figure 4: Locations of drainage divides with high (orange) and low (blue) asymmetry by geomorphic metric, where divide line thickness increases with divide Strahler order (4-10). White divides are not classified as having either high or low asymmetry. Panels include CRR (a), HSG (b), $k_{snP}$ (c) and $\chi_P$ (d). Metric-specific thresholds for high and low DAI can be found in Table S1. Numbers in (d) correspond to low-relief landscape features labelled in Fig. 2d. See Fig. S3 for increased visibility of high and low asymmetry in low order divides.**

*"L289-294: As this is not a new insight in general terms (e.g., it's a central point of Forte & Whipple, 2018, among other papers), it would be good to add citations to indicate as such."*

We agree and added the following references to following this statement (lines 293-294): "The integral metrics can thus reflect transient processes in distal parts of the catchment, rather than processes local to a divide (Whipple et al., 2017; Forte and Whipple, 2018)."

*"L296-297: Did you mean to cite Adams here? It's not clear how that paper is relevant to the point you're making?"*

Yes, we thank the reviewer for catching this mistake and have removed the reference in the revised manuscript.
* * *
*"L319-324: This all makes sense, but the extent to which this is or is not a problem within your datasets are hard to assess. I.e., while it's certainly true that a particular metric on one (or both) side(s) of the divide effectively reaching its threshold would lead to underestimates of what the "true" DAI should be, this is only relevant if the metrics are in the right range, no? While this is a bit challenging to know a priori since there are not single global values of what appropriate thresholds for each metric are and it's not unreasonable to assume that some (or maybe even many) metrics may be near or at threshold given the tectonic activity of the region, it would be good to have some assessment of whether many (or any) of the raw values of the chosen metric display a threshold like behavior. I.e., if you just plotted all hillslope gradients on a histogram, do you see a distribution that's reflective of many values being at/near a suite of thresholds?"*

The reviewer's suggestion of a histogram of hillslope values to confirm the existence of threshold behavior in this metric is an excellent one. Below we show a new supplemental figure we have added which shows a steep decline in the number of divides with hillslope gradients (HSG) above 30°, consistent with threshold behavior. Note that the y-axis is in log-scale. This figure has been added to the Supplement (Fig. S5).

[Figure]

**Figure S5. Histogram showing the distribution of hillslope gradient (HSG) values in the HDM. The data indicate an abrupt decline in the frequency of HSG values above ~30°, suggesting that hillslopes in the study region may reach a threshold steepness around this point.**

We also added a reference to Liu et al. (2021b) who determined the threshold hillslope value for each catchment in the HDM (Salween, Mekong, Yangtze, Yalong, Dadu, and Min rivers) from their mode hillslope values. They measured mode values ranging from 26° (Mekong River) to 33° (Dadu River) and subsequently propose a threshold HSG value of 30 ± 5° for the HDM. We have added the following sentences to the manuscript in the paragraph spanning lines 319-324: "The histogram of HSG values in the HDM (Fig. S5) shows a marked decline in frequency above a slope of 30° in the HDM, indicating that hillslopes are reaching a threshold. This supports the 30 ± 5° HSG threshold determined by Liu et al. (2021) based on the mode slope values of major catchments in the HDM."

*"L324-325: Even without the context of thresholds, this seems prudent as it's not clear from first principles that a particular DAI based on different metrics would be expected to lead to the same rate of divide migration."*

Yes, exactly. We agree with this comment and, as it was not explicitly stated in the original manuscript, have modified the text from "Due to this potential for threshold behavior, we use metric-specific thresholds to distinguish drainage divides with high and low asymmetry" to "Due to this potential for threshold behavior and the expectation that a given DAI value corresponds to a different erosion rate in each metric, we use metric-specific thresholds to distinguish drainage divides with high and low asymmetry."

*"References cited in this review:*

*Adams, B. A., Whipple, K. X., Forte, A. M., Heimsath, A. M., & Hodges, K. V. (2020). Climate controls on erosion in tectonically active landscapes. Science Advances, 6(42). https://doi.org/10.1126/sciadv.aaz3166*

*Forte, A. M., & Whipple, K. X. (2018). Criteria and tools for determining drainage divide stability. Earth and Planetary Science Letters, 493, 102–117. https://doi.org/10.1016/j.epsl.2018.04.026*

*Leonard, J. S., & Whipple, K. X. (2021). Influence of Spatial Rainfall Gradients on River Longitudinal Profiles and the Topographic Expression of Spatially and Temporally Variable Climates in Mountain Landscapes. Journal of Geophysical Research: Earth Surface, 126(12). https://doi.org/10.1029/2021JF006183*

*Leonard, J. S., Whipple, K. X., & Heimsath, A. M. (2023). Isolating climatic, tectonic, and lithologic controls on mountain landscape evolution. Science Advances, 9(3), eadd8915. https://doi.org/10.1126/sciadv.add8915*

[revised manuscript text omitted]

---

## Author Comment (AC2)

**Response to Anonymous Referee #2**

*"General comments*

*This work investigated drainage divide asymmetry of the Hengduan Mountains (HDM) using four geomorphic metrics (CRR, HSG, ksnP, and χP) to understand the spatial and temporal patterns of geometric transience of river network in this region. They find clear evidence of widespread transience through a high incidence of strongly asymmetrical divides throughout the HDM with the four geomorphic metrics. The study of landscape transient effect in this complex area is challenging, and the work provides such test using four metrics, and compare the difference between these metrics. From these new aspects, the manuscript is suitable for publishing in ESurf. However, the manuscript may have some problems on the geomorphic metrics that need to be addressed and which may require substantial revision.*

*(1) ksn and χ are both precipitation-corrected to account the strong precipitation gradient in this region in equations (4) and (6), but the authors seem use the 'local' mean annual precipitation P, in fact it should be the 'upstream' mean annual precipitation of a reference point, i.e., rainfall rate averaged over A, according to Adams et al. (2020). Local and upstream average are quite different."*

We are glad that the reviewer mentioned this point because, while we do use the upstream mean annual precipitation when calculating precipitation-corrected $k_{sn}$ and χ, this was not stated in the text. To address this, we have added the blue text to line 181: "…we multiply $A$ by the upstream mean annual precipitation (P)…."

*"For calculating the χ and χP, the authors chose a base-level of 500 m for the study area to approximate the elevation at the western edge of the Sichuan Basin. This is fine for a base-level of most streams of the Yangtze River in the HDM. However, the work mainly studies the transience of the HDM in the Three Rivers region (not only the Yangtze River), which have three different outlets. Whether the results and conclusions are sensitive if the base-level elevations are set to 1000 m or 1500m, which are approximately at the plateau margin."*

The reviewer is correct that the base-level for the χ and χ$_P$ integration matters and has to be carefully chosen. χ is computed relative to a local base level, where perturbations caused by rock uplift occur only above this base level. Therefore, the stability of the base level's elevation is crucial. We chose the elevation of the Sichuan basin as the base level due to its tectonic inactivity, assuming that its elevation remains constant. Similarly, for the Salween and Mekong rivers, situated at an elevation of 500 meters, they already traverse regions characterized by tectonic inactivity, possibly flood plains.

Of the three main rivers in the study region (Salween, Mekong, Yangtze), the Yangtze River where it enters the Sichuan basin, has the highest base-level. Utilizing the a lower base-level from one of the other rivers would lead to the inclusion of alluvial reaches for rivers draining to the Sichuan basin. This would influence the χ and χ$_P$ integration through distinctly higher erodibility and non-detachment-limited conditions in the unconsolidated sediments of the Sichuan Basin. Moreover, as mentioned in the text (line 193) and shown in Figure 1, the Yangtze River drainage covers a majority of the HDM surface area. With the exception of the Salween and Mekong catchments, all streams in the analysis region eventually drain into the Yangtze and through the Sichuan Basin.

Choosing a higher base-level, as suggested by the reviewer, would change the resulting χ and χ$_P$ maps and some across-divide differences, but would also substantially reduce the area of analysis. Choosing a higher base-level would only be necessary if there were an elevation-dependent variation in the $U/K$-ratio that should be removed from the analysis. We acknowledge this limitation of χ in the text (lines 62-69 and 292-293) and capitalize on it to test for such $U$ and $K$ variations by comparing crossdivide differences in $\chi$ to the other geomorphic metrics, which are only sensitive to local erosion rate (explained on lines 88-90, 289-296, 327-330, and 436-440, and visualized in Figure 5). As stated on lines 330-333, we found that "divide migration direction inferred by $\chi_P$ still agrees with local metrics a majority of the time, especially in strongly asymmetrical divides. This suggests that, despite several areas of localized uplift and heterogeneous lithology across the HDM, changes in uplift and erodibility are small enough, or slow enough, that $\chi_P$ values at the divides remain dominated by the length of the rivers and their area distributions in most places."

To further address the reviewer comment, we have also prepared two additional $\chi$ maps, one with a 1000 m base-level and one with a 1500 m base-level, to show the resulting reduction in analysis area. DAI was calculated for three divide segments in the HDM at each base-level and compared (Table R1; black circles in Fig. R1). These are shown below next to the $\chi$ map with the original 500 m base-level, but not included in the revised manuscript:

[Figure]

**Figure R1. Series of $\chi$ maps for the HDM calculated from different base-levels. Left map shows $\chi$ from a 500 m base-level (as used in this study), center map shows $\chi$ from a 1000 m base-level, and right map shows $\chi$ from a 1500 m base-level. Color ramps are set to the same minimum and maximum values for all maps. Black circles indicate locations where divide asymmetry was measured to show how DAI changes with base-level. From west to east, these circles are located at the Salween River-Mekong River divide, Yangtze River-Yalong River divide, and Dadu River-Anning River divide. Their DAI values are compared in the Table R1 (below).**

DAI for the divide between the Salween and Mekong rivers is near constant at all three base-levels. DAI for the divide between the Yangtze and Yalong Rivers gradually increases with base-level, likely because of the removal of more of the steep, narrow portion of the Yalong River. DAI for the divide between the Dadu and Anning rivers is roughly the same at 500 m and 1000 m, but decreases by half at 1500 m because this base-level is higher than the knickpoint in the Anning River associated with the capture of its headwaters by the Dadu River. These examples demonstrate that while, yes, changing the base-level can alter cross-divide differences in $\chi$, ultimately there is no one base-level that can eliminate transients and U/K variations in the HDM. Overall, the value of $\chi$ as an interpretation tool remains.

**Table R1. Comparison of DAI values calculated from $\chi$ with three different base-levels for three divide segments in the HDM. The locations of the divide segments are shown in Fig. R1.**

| | | DAI Value | | |
|---|---|---|---|---|
| **Map Location** | **Divide Location** | **500 m Base-level** | **1000 m Base-level** | **1500 m Base-level** |
| Western-most (left) | Salween River – Mekong River Divide (in Three Rivers Area) | 0.15 | 0.19 | 0.15 |
| Central | Yangtze River – Yalong River Divide | 0.37 | 0.46 | 0.63 |

| | (at Yanyuan Basin) | | | |
|---|---|---|---|---|
| Eastern-most (right) | Dadu River – Anning River Divide (at "main" windgap) | 0.42 | 0.44 | 0.21 |

For all the above-mentioned reasons, we decided to keep the base-level at 500 m. However, for additional clarity, we will modify the following passages (blue text is new):

L192-194: "A base-level of 500 m was used for the study area to approximate the elevation at the western edge of the Sichuan Basin. The Sichuan Basin is a part of the stable South China Tectonic Block and serves as a natural base-level for most streams in the HDM via the Yangtze River, which possesses the highest base-level of any major river in the region (Fig. 1a)."
* * *
*"Lines 11-12: The authors claim that they evaluate the relative time scales of this transience by comparing drainage divide asymmetry, but I did not see any time scales of transience in this work."*

$\chi$ and $\chi_P$ are metrics calculated by integrating drainage area (or a relative discharge proxy) for the entire drainage basin. Therefore, these metrics tend to record transience on long time-scales. In contrast, metrics calculated locally at the divide, such as hillslope gradient, are proxies for the short time scale, even instantaneous, divide motion. This is expressed in the introduction of the original manuscript (lines 66-70), the results (lines 290-291), the conclusion (lines 436-437) and the discussion (lines 315-318) from which we cite: *"Metrics are computed on different spatial and, consequently, temporal scales; for example, $\chi_P$ is integrated across entire catchments and represents long-term trends in landscape evolution (Beeson et al., 2017; Scheingross et al., 2020), while HSG is averaged across individual divide segments and adjusts to local, short-term changes to catchment structure."* To improve the clarity of the text, we have modified the underlined text so it now reads "…while HSG is averaged across individual divide segments and reflects local, short-term rates of erosion."
* * *
*"Lines 88-90: There are many one-sentence paragraphs in the manuscript, it is quite strange to have one-sentence paragraphs, try to minimize them."*

The sentence pointed out in the comment has been added to the previous paragraph. We have also gone through the manuscript and either incorporated the other one-sentence paragraphs into their preceding paragraphs (lines 183-184) or expanded the paragraph into multiple sentences (lines 162-163).
* * *
*"Lines 178-179: The authors refer to Fig. S2 "A best-fit θref of 0.45 was determined for the HDM through Bayesian optimization with the mnoptim function in Topotoolbox", but Fig. S2 is not on the river concavity, missing a figure?"*

We thank the reviewer for catching this oversight. We have added the figure (shown below) to the revised version and have updated the figure numbers accordingly.

[Figure]

**Figure S2. Results of Bayesian optimization of reference concavity ($\theta_{ref}$) for the HDM calculated using the *mnoptim* function in Topotoolbox. Best-fit $\theta_{ref}$ of ~0.45 ("model minimum feasible") is marked with a red star. White circles mark calculated $\theta$ (*m/n*) values and their corresponding estimated objective function values for each of the 100 model iterations. The model mean is shown with a solid red line and its corresponding error is shown with a dashed red line. The dotted grey line is the noise error bars.**
* * *
*"Lines 183-184 and Fig. S1: Maybe put ksn and ksnP, χ and χP together for easy of comparing? One-sentence of paragraph is not necessary here."*

The reviewer makes an interesting suggestion that we considered. However, we determined that the spatial variations between these metrics are too minor for visual comparison. As Fig. 4 shows, the correlation coefficient between $k_{sn}$ and $k_{snP}$ is 0.99 and between $\chi$ and $\chi_P$ is 0.97. However, we have chosen to add an additional figure suggested by the reviewer in a later comment, which compares where high and low asymmetry divides differ between $k_{sn}$ and $k_{snP}$ and between $\chi$ and $\chi_P$ to the supplement, which we believe suits a similar purpose (see response to later comment for more details).

As mentioned in response to a previous comment, we added the sentence to the previous paragraph to avoid having a one-sentence paragraph.
* * *
*"Lines 238-240: It seems that the threshold for "high" (95th percentile) and "low" (5th percentile) divide asymmetry in each metric is chosen arbitrary?"*

The reviewer is correct that the 5th and 95th percentile thresholds are somewhat arbitrary. However, these are commonly used thresholds for outliers in statistical analysis and are relatively conservative.

We chose to use percentiles over alternative methods, such as 1.5x the interquartile range (IQR), because this approach allows for consistency across the different metrics, providing a uniform method of classification that is less dependent on their individual DAI distributions. This is important because, as shown in figures 3 and S4, in Table S1, and mentioned on lines 266-270, DAI distributions are right-skewed and vary by metric. If we were to instead use 1.5x IQR to define high and low divide asymmetry, no metric would have low asymmetry divides and all except local relief (LR) would have more high asymmetry divides (see table below).

**Table comparing outlier thresholds using the percentile and 1.5x IQR methods for drainage divide asymmetry by magnitude (DAI) by metric.**

| Metric | Min | Max | Percentile | | IQR | |
|---|---|---|---|---|---|---|
| | | | 5th | 95th | Lower | Upper |
| CRR | 1.80E-05 | 0.9674 | 0.0040 | 0.1623 | 0 | 0.1121 |
| LR | 1.10E-05 | 0.9674 | 0.0033 | 0.1004 | 0 | 0.1397 |
| HSG | 1.00E-06 | 0.6057 | 0.0054 | 0.1984 | 0 | 0.1612 |
| ksn | 4.40E-05 | 1.0000 | 0.0155 | 0.5609 | 0 | 0.5066 |
| ksnP | 1.00E-05 | 1.0000 | 0.0152 | 0.5641 | 0 | 0.5077 |
| χ | 8.00E-06 | 0.6878 | 0.0052 | 0.2180 | 0 | 0.1880 |
| χP | 7.00E-06 | 0.6448 | 0.0048 | 0.2077 | 0 | 0.1626 |

*"Lines 243-249 and Fig. S2: The authors explain the difference of highly asymmetric drainage divides between χ and χP, but I did not understand by looking at the Fig. S2. It is better to show an overlap figure marking the difference between metrics χ and χP, and a figure marking the difference between metrics ksn and ksnP."*

We thank the reviewer for this suggestion. Though, as mentioned in a previous response, the differences between these metrics are very small, such an overlap figure is an excellent way to highlight the minor spatial differences between them. We include the new figure below and have added it to the supplement in the revised manuscript.

[Figure]

**Divide Order**
4 ———— 10

1 metric has high or low asymmetry

2 groups share    all 3 groups share

**Figure S4: Locations of drainage divides that have do not have consistently high or low asymmetry between two similar geomorphic metrics (i.e., one metric has high asymmetry when the other does not) are marked in red. Divide line thickness increases with divide Strahler order (4-10). Panels include (a) CRR vs. LR, in which 2,533 divide segments (11%) have conflicting asymmetry classifications, (b) $k_{snP}$ vs. $k_{sn}$, in which 1,313 divide segments (6%) have conflicting asymmetry classifications, and (c) $\chi_P$ vs. $\chi$, in which 1,599 divide segments (7%) have conflicting asymmetry classifications. In no instance in a-c does any metric have low asymmetry when its counterpart has high asymmetry. Panel (d) shows the locations of divide segments for which all of the metric pairs in a-c have conflicting asymmetry classifications (pink, 11 divides) and of divide segments for which two of the metric pairs have conflicting asymmetry classifications (orange, 342 divides). Metric-specific thresholds for high and low DAI can be found in Table S1.**

*"Fig. 3. It is very difficult to understand this figure. It took much of my time to understand it. I suggest simplifying this figure. For example, changing the y-axis in right-hand side to the grey color. Move '%' in y-axis of the left-hand side to the end of Migration direction agreement."*

We like the reviewer's suggestions on how to improve Fig. 3 and have updated it, and its counterpart in the supplement, accordingly. Below we show the revised Fig. 3:

[Figure]

**Figure 3: Plots of percent agreement in divide migration direction between chosen metrics and all calculated metrics (colored points), binned by corresponding divide asymmetry index (DAI) for indicated metric in intervals of 0.05. Grey histograms show the distributions of DAI values in log-scale for each metric. Higher DAI corresponds with increased agreement in migration direction between metrics. Histograms show variability in DAI distributions in different metrics.**
* * *
*"Check the unit of KsnP, the unit of Ksn is m^0.9, but KsnP has considered the precipitation with unit of m/yr."*

The reviewer is correct that the unit of $k_{snP}$ is not the same as $k_{sn}$ because of the inclusion of precipitation in the equation and we are very glad that they caught this mistake. Following Eq. 4,

$$k_{snP} = (AP)^{\theta_{ref}} S \,,$$

since slope ($S$) is unitless, the unit of $k_{snP}$ is

$$\left(m^2 \cdot \frac{m}{a}\right)^{0.45} = m^{1.35} a^{-0.45} \,.$$ This has been corrected in the manuscript text and figures.
* * *
*"All equations require a common or a point in the end, they are currently missing throughout the manuscript."*

We thank the reviewer for pointing out this oversight. The text has been updated accordingly.
* * *
*"I hope that these comments are helpful for the revision."*

We thank the reviewer again for their constructive critique and helpful suggestions.

---

## Author Response (AR2)

**Response to the Editor**

*"Minor comments:*

*Reviewer 2 rightly pointed out that the chosen base level for calculation of χ and χP has an influence on the calculated divide asymmetry index. In your response to reviewer 2, you presented some analyses which show the difference in DAI with base-levels of 1000 and 1500 m. However, none of this analysis or discussion has appeared in the revised manuscript. I think this is an important point, and urge you to highlight this more clearly in your revised manuscript or at least in your supplemental information."*

We acknowledge the critical role base-level plays in the calculation of $\chi$ and $\chi_P$. It is this base-level sensitivity that necessitates that the primary criterion for selecting base-level is a tectonically stable reference point, with all upstream area responding to uplift with respect to this stationary elevation. Stationarity of the base-level elevation requires that downstream reaches of all rivers have the same steepness, so the same uplift and incision rates. The Sichuan Basin is a rigid tectonic block that has served as a temporally stable base-level for more than two-thirds of the rivers in the HDM throughout the Cenozoic. The lower reaches of the remaining streams (belonging to the Salween and Mekong catchments) are also characterized by tectonic inactive areas and thus have low channel steepnesses with little variation in gradient. By selecting an elevation of 500 meters for our baselevel, we are effectively using the Sichuan Basin and the upper boundary of the low-steepness Mekong and Salween River regions, removing any large differences in $\chi$ arising from these long, flat reaches (Yang et al., 2020; Fox et al., 2020; Jiao et al., 2022; Pan et al., 2023).

Furthermore, most cross-divide comparisons in the study are internal to a single river catchment (i.e., Yangtze, Mekong, or Salween), so base-level plays no role (Yang et al., 2015). Only at the borders between the Yangtze, Mekong, and Salween rivers is the chosen base-level important. As shown in our response to reviewer 2 (Fig. R1, Table R1), despite minor differences in asymmetry magnitudes at higher base-levels, there are no striking differences in spatial patterns of $\chi$ within or between the major river catchments at higher base-levels. This indicates that the applied base-level of 500 m successfully removes tectonically inactive portions of the drainage network, while maximizing the area of analysis.

We therefore chose not to add these maps to the revised manuscript or supplement. We believe that including multiple $\chi$ maps with varying base-levels adds unnecessary ambiguity that may confuse readers and suggest that base-level is arbitrarily chosen, rather than carefully set based on established criteria.

*"In Line 394 you state that the general agreement between χP and the other metrics suggests that changes in uplift or erodibility are localised or slow enough that χP acts as a good indication of divide migration direction. However, there are many divides where there are significant differences between χP and the other metrics, and indeed you show that χP is the most different to the other metrics in Figure 3. One way to tackle the erodibility part of the problem would be to include geological data from the region, and test whether the divides with disagreement between χP and the other metrics are those with more variability in catchment lithology/erodibility."*

The editor is correct that $\chi_P$ shows the lowest agreement on migration direction with the other metrics. However, we would like to note that the difference to the other metrics is minor despite

comparing a catchment-wide integration variable ($\chi_P$) to the metrics calculated locally at the divide. For example, $\chi_P$ agrees on migration direction with $k_{snP}$ a similar amount as HSG does at 59.3% and 59.9%, respectively (Fig. 3). $\chi_P$ and HSG also have similar levels of agreement on migration direction with CRR at 64.5% and 68.1%, respectively. The worst agreement between $\chi_P$ and another metric is still 57% (with HSG). Given the complex geologic and tectonic setting of the HDM, we find this behaviour quite remarkable.

To address the problem of divide metrics being influenced by lithology, we have added a lithologic map to the supplement of the revised manuscript (Fig. S1, below).

In this figure, one can see that the strong heterogeneity in lithology persists throughout the entire HDM, without any clear spatial patterns that correlate with strongly asymmetric divides or where $\chi_P$ and local-scale metrics disagree on migration direction. Given that $\chi$ metrics depend on the integral of erodibility, the effect tends to be small unless there are large-scale, regional differences in erodibility.

[Figure]

**Figure S1. Spatial comparison of HDM lithology and $\chi_P$ asymmetry magnitude and agreement with local-scale metrics shows no clear patterns between lithology and strong divide asymmetry or migration direction disagreement. Lines represent drainage divides with line thickness corresponding to $\chi_P$ DAI. Divide segments where migration directions in $\chi_P$ and a majority of local-scale metrics (2+/3) agree are white; segments where $\chi_P$ indicates a divide migration direction contrary to 2 or more local-scale metrics are black. Lithology abbreviations are ice/water (IW), unconsolidated sediments (UN), pyroclastic rocks (PY), evaporites (EV), acidic plutonic rocks (PA), intermediate plutonic rocks (PI), basic plutonic rocks (PB), carbonate sedimentary rocks (SC), mixed sedimentary rocks (SM), siliciclastic sedimentary rocks (SS), metamorphic rocks (MT), acidic volcanic rocks (VA), intermediate volcanic rocks (VI), and basic volcanic rocks (VB). Lithologic data are from the GLiM global lithologic map (Hartmann and Moosdorf 2012a).**

*"Line 180: although it is good to see that the best fit concavity of 0.45 was tested rather than assumed, it would be beneficial to explain how the mnoptim function actually works*

*to improve clarity for readers of the manuscript. Does this use a disorder method to test for best fit concavity or does it look for collapse of tributaries onto the main stem?"*

To better explain how the 'mnoptim' function works we added the following sentence to the methods:

"The *mnoptim* function loops through subsets of the drainage network, determines in each iteration the concavity that best linearizes the stream profiles of the subset, and then tests this concavity on the remainder of the network to find the best-fit for the entire drainage network."
* * *
*"Line 186: It's not clear to me what you mean by "critical drainage area threshold" here. Do you mean A0 in the calculation of χ? This should be more clearly explained and justified."*

The critical drainage area, refers to the drainage area required for stream initiation. We have clarified this in the revised manuscript. We cite from the revised sentence:

"For all calculations, a critical drainage area threshold for stream initiation of 5 km$^2$ was used."
* * *
*"Line 289: this point could be made stronger by including a quantification of "majority" e.g. state the percentage of divide segments that show agreement. There are a couple of other points through the text where additional quantification instead of stating simply "majority" could also help."*

We thank the editor for this comment and have added the percentage of divides that show agreement to the referenced sentence and others that did not explicitly state percentages. We cite from the revised manuscript:

L291: "Inferred divide migration directions agree between geomorphic metric pairs for a majority (57% or more) of the 22,837 divide segments analyzed."

L344: "Overall, geomorphic metric agree on divide migration direction in a majority of cases (Figs. 3 and 7), with total agreement between any two metrics ranging from 57% to 97%."

L366: "However, divide migration direction inferred by $\chi_P$ still agrees with at least two of the three local metrics 69.2% of the time, especially in divides with strong $\chi_P$ asymmetry (83.7%)."
* * *
*"Best wishes,*
*Fiona Clubb (Associate Editor)"*

**References Cited:**

Fox, M., Carter, A., and Dai, J.-G.: How Continuous Are the "Relict" Landscapes of Southeastern Tibet?, Frontiers in Earth Science, 8, 2020.

Hartmann, J. and Moosdorf, N.: The new global lithological map database GLiM: A representation of rock properties at the Earth surface, Geochemistry, Geophysics, Geosystems, 13, https://doi.org/10.1029/2012GC004370, 2012.

Jiao, R., Fox, M., and Yang, R.: Late Cenozoic erosion pattern of the eastern margin of the Sichuan Basin: Implications for the drainage evolution of the Yangtze River, Geomorphology, 398, 108025, https://doi.org/10.1016/j.geomorph.2021.108025, 2022.

Pan, R., Han, Z., Su, Q., Li, G., Li, X., Li, Y., and Wang, X.: Two-staged uplift of the southeast margin of the Tibetan plateau revealed by river longitudinal profiles, Front. Earth Sci., 11, https://doi.org/10.3389/feart.2023.1124362, 2023.

Yang, R., Willett, S. D., and Goren, L.: In situ low-relief landscape formation as a result of river network disruption, Nature, 520, 526–529, https://doi.org/10.1038/nature14354, 2015.

Yang, R., Suhail, H. A., Gourbet, L., Willett, S. D., Fellin, M. G., Lin, X., Gong, J., Wei, X., Maden, C., Jiao, R., and Chen, H.: Early Pleistocene drainage pattern changes in Eastern Tibet: Constraints from provenance analysis, thermochronometry, and numerical modeling, Earth and Planetary Science Letters, 531, 115955, https://doi.org/10.1016/j.epsl.2019.115955, 2020.